

# Insights into nitrogen fixation below the euphotic zone: trials in an oligotrophic marginal sea and global compilation

Siqi Wu[1,2], Moge Du[1,2], Xianhui Sean Wan[1,3], Corday Selden[4], Mar Benavides[5], Sophie Bonnet[5], Robert Hamersley[6], Carolin R. Löscher[7,8], Margaret R. Mulholland[4], Xiuli Yan[9], Shuh-Ji Kao[1,2,10]

[1]State Key Laboratory of Marine Environmental Science, Xiamen University, Xiamen, 361102, China
[2]College of Ocean and Earth Sciences, Xiamen University, Xiamen, 361102, China
[3]Department of Geosciences, Princeton University, Princeton, NJ 08544, USA
[4]Department of Ocean & Earth Sciences, Old Dominion University, Norfolk, VA 23529, USA
[5]Aix Marseille Univ., Université de Toulon, CNRS, IRD, MIO UM 110, Marseille, 13288, France
[6]Environmental Studies, Soka University of America, Aliso Viejo, 92656, USA
[7]Nordcee, Department of Biology, University of Southern Denmark, Odense, DK-5230, Denmark
[8]D-IAS, University of Southern Denmark, Odense, DK-5230, Denmark
[9]Marine Science Institute and Guangdong Provincial Key Laboratory of Marine Biotechnology College of Science, Shantou University, Shantou, 515063, China
[10]State Key Laboratory of Marine Resources Utilization in South China Sea, Hainan University, Haikou, 570228, China

*Correspondence to*: Shuh-ji Kao (sjkao@xmu.edu.cn)

**Abstract.** Nitrogen ($N_2$) fixation, the energetically expensive conversion of $N_2$ to ammonia, plays an important role in balancing the global nitrogen budget. Defying historic paradigms, recent studies have detected non-cyanobacterial $N_2$

fixation in deep, dark oceanic waters. Even low volumetric rates can be significant considering the large volume of these waters. However, measuring aphotic $N_2$ fixation is an analytical challenge due to the low particulate nitrogen (PN) concentrations. Here, we investigated $N_2$ fixation rates in aphotic waters in the South China Sea (SCS). To increase the sensitivity of $N_2$ fixation rate measurements, we applied a novel approach requiring only 0.28 μg N for measuring the isotopic composition of particulate nitrogen. We conducted parallel [15]$N_2$-enriched incubations in ambient seawater, seawater

amended with amino acids and poisoned ($HgCl_2$) controls, along with incubations that received no tracer additions to distinguish biological $N_2$ fixation. Experimental treatments differed significantly from our two types of controls, those receiving no additions and killed controls. Amino acid additions masked $N_2$ fixation signals due to the uptake of added [14]N-amino acid. Results show that the maximum dark $N_2$ fixation rates ($1.28 \pm 0.85$ nmol N $L^{-1}$ $d^{-1}$) occurred within upper 200 m, while rates below 200 m were mostly lower than 0.1 nmol N $L^{-1}$ $d^{-1}$. Nevertheless, $N_2$ fixation rates between 200 and 1000 m

accounted for $39 \pm 32$ % of depth-integrated dark $N_2$ fixation rates in the upper 1000 m, which is comparable to the areal nitrogen inputs via atmospheric deposition. Globally, we found that aphotic $N_2$ fixation studies conducted in oxygenated environments yielded rates similar to those from the SCS (< 1 nmol N $L^{-1}$ $d^{-1}$), regardless of methods, while higher rates were occasionally observed in low-oxygen (< 62 μM) regions. Regression analysis suggests that particulate nitrogen concentrations could be a predictive proxy for detectable aphotic $N_2$ fixation in the SCS and eastern tropical south Pacific.





Our results provide the first insight into aphotic N₂ fixation in SCS and support the importance of the aphotic zone as a globally-important source of new nitrogen to the ocean.

## 1 Introduction

The marine fixed nitrogen (N) inventory is primarily determined by the input of new N via atmospheric deposition, biological dinitrogen ($N_2$) fixation, riverine flux and its loss from the ocean via denitrification and anammox (Wang et al.,
2019). N inputs and losses in the global ocean appear to have been balanced over the last 10 ka (Gruber, 2004), however, budgets determined from bottom-up measures have often shown discrepancies due to our developing understanding of the N cycle in the ocean. The marine N budget is a fundamental question in marine biogeochemistry, since it affects marine productivity and thus carbon dioxide ($CO_2$) sequestration by the ocean (Codispoti, 2007). However, many parts of the N budget remain controversial (Codispoti, 2007; Eugster and Gruber, 2012; Gruber and Galloway, 2008; Wang et al., 2019;
Zehr and Capone, 2020). $N_2$ fixation, an important external source of new N to the open ocean (Jickells et al., 2017), and was long assumed to occur primarily within the euphotic zone where photoautotrophy can accommodate the high energetic cost of $N_2$ fixation (Zehr and Kudela, 2011). However, the observations of pervasive *nifH*-gene containing non-photosynthetic marine microorganisms (Moisander et al., 2008; Zehr et al., 1998) highlighted the potential for non-cyanobacterial $N_2$ fixation, motivating broader investigations into the range of diazotrophs in the ocean, including the mesopelagic zone
(Moisander et al., 2017).

Recently, a number of studies detected $N_2$ fixation in aphotic zone (aphotic $N_2$ fixation, ANF) across a wide range of marine regimes including both hypoxic (Bonnet et al., 2013; Chang et al., 2019; Dekaezemacker et al., 2013; Farnelid et al., 2013; Fernandez et al., 2011; Gradoville et al., 2017; Selden et al., 2019; Hamersley et al., 2011; Loescher et al., 2014; Löscher et al., 2016) and oxic waters (Benavides et al., 2015, 2016, 2018a; Gradoville et al., 2017; Mulholland et al., 2019; Rahav et al.,
2013; Weber, 2015). However, reported rates are highly variable, ranging from below the detection limit to up to 35.9 nmol N L⁻¹ d⁻¹ in the oxygen minimum zone in the Eastern Tropical North Pacific (ETNP) (Selden et al., 2019). Among reported rates, ANF, even if low (mostly below 1 nmol N L⁻¹ d⁻¹), have been shown to contribute 6-100 % of water column integrated $N_2$ fixation rates (Benavides et al., 2018a, 2016, 2015; Bonnet et al., 2013; Farnelid et al., 2013; Fernandez et al., 2011; Hamersley et al., 2011; Rahav et al., 2013). This large contribution is mostly due to the large volume of aphotic water and
the assumption that ANF can be integrated trapezoidally (Benavides et al., 2018b; Zehr and Capone, 2020). However, owing to the low rates of ANF and low concentrations of PN in the ocean's interior, extreme care must be taken to assure that reported rates are robust and above the limits of analytical detection (Gradoville et al., 2017; Moisander et al., 2017; White et al., 2020). ANF measurement biases could be caused by insufficient enrichment with ¹⁵N₂, processes other than N₂ fixation (e.g. isotopic fractionation during DIN or DON assimilation), and inflated errors by elemental analysis–isotope ratio mass
spectrometry (EA-IRMS) at low PN mass. Thus, methods with sufficient ¹⁵N₂ enrichment together with controls and sophisticated isotopic measurements are crucial. Overall, the relatively few rate measurements, their variability, and the fact



that many observations are at or near the limits of analytical/computational detection lead to large uncertainties in the contribution of ANF to the global fixed N budget, thus, further investigations on ANF in different regions and its controlling factors are needed.

Given the high energetic cost of $N_2$ fixation relative to assimilation of nitrate (the major form of dissolved inorganic nitrogen (DIN) at depth; Zehr and Kudela, 2011; Falkowski, 1983), it seems counter-intuitive that this process occurs in nitrate-replete and energy-starved aphotic waters (Moisander et al., 2017). Several hypotheses have been proposed to explain the occurrence of ANF: (1) Deep sea diazotrophs lack the transporters necessary to assimilate or reduce DIN (Bombar et al., 2016; Karl et al., 2002), leaving $N_2$ fixation as the only available source of N, which is the case for some diazotrophic

cyanobacteria (Caputo et al., 2018; Hilton et al., 2013); (2) $N_2$ fixation helps cells maintain an ideal intracellular redox state (Bombar et al., 2016); (3) DIN could be depleted in aggregates as a result of microbial succession, in densely packed microbial consortia (Bombar et al., 2016; Dekas et al., 2009); (4) Micro-anerobic environments within particles or lower ambient oxygen concentrations could be a suitable niche for diazotrophs because nitrogenase can be inactivated by oxygen (Postgate, 1970; Paerl and Prufert, 1987; Paerl and Carlton, 1988; Paerl and Bebout, 1988; Farnelid et al., 2019; Pedersen et

al., 2018).

In addition, since non-cyanobacterial diazotrophs are most likely heterotrophic, the energy required for $N_2$ fixation may be supplied by labile dissolved organic matter (DOM) (Bombar et al., 2016), as for other deep-sea, aphotic microbes (Arístegui et al., 2009). Accordingly, a set of labile DOM compounds, including dissolved free amino acids (DFAA) (Benavides et al., 2015; Bonnet et al., 2013; Rahav et al., 2013, 2015; Selden et al., 2019), glucose (Dekaezemacker et al., 2013; Gradoville et

al., 2017; Loescher et al., 2014; Löscher et al., 2016; Selden et al., 2019), carbohydrate mixtures (Benavides et al., 2015; Bonnet et al., 2013), ATP (Bonnet et al., 2013), and pure xanthan gum from *Xanthomonas campestris* (Rahav et al., 2013) have been used in enrichment experiments to examine DOM limitation of ANF. Although most studies reported ANF stimulation by DOM additions, the effect varied between regions and DOM sources. DFAA additions resulted in one to seven-fold enhancement of ANF in most studies (Benavides et al., 2015; Bonnet et al., 2013; Rahav et al., 2013, 2015;

Selden et al., 2019), although some studies also reported no enhancement (Benavides et al., 2015; Selden et al., 2019), or even an inhibitory effect (Selden et al., 2019). These inconsistencies could be due to population-specific substrate preferences and metabolic diversity, variability in energy or carbon limitation, decreased DOM utilization under low temperature (Selden et al., 2019), or incubation artifacts (e.g., pressure, bottle effects, etc). DOM addition experiments remain sparse, and further experiments in different regions would be helpful to elucidate the effect of DOM on ANF.

Here, we present the first reported $N_2$ fixation measurements from the mesopelagic SCS, the largest oxygenated marginal sea in the western Pacific Ocean. To better constrain low $N_2$ fixation rates below the euphotic zone, we first utilized alkaline persulfate digestion (Knapp et al., 2005) coupled with a denitrifier method (persulfate-denitrifier method hereafter) to measure PN concentration and isotopic composition (Casciotti et al., 2002; McIlvin and Casciotti, 2011; Sigman et al., 2001). This method requires only 0.28 µg N (or 20 nmol N) rather than the roughly 10 µg N required when using standard

EA-IRMS analysis (White et al., 2020). We conducted control experiments to account for all possible sources of error





(signals other than ANF), to estimate ANF rates. We also tested whether DFAA additions stimulate ANF in the SCS and, in so doing, identified potential problems inherent to such substrate addition experiments.

## 2 Methods

### 2.1 Field sampling

Water samples were collected from 4 stations during 2 cruises in the SCS aboard the *R/V Tan Kah Kee* in August 2018 (KK1806) and July 2019 (KK1905; Fig. 1). Salinity, temperature, dissolved oxygen (uncalibrated, performed to observe vertical trends), fluorescence, and photosynthetically active radiation (PAR) data were obtained via a SeaBird conductivity-temperature-depth (CTD) profiler (SBE 911 plus). Water samples for incubations were obtained from a rosette of $24 \times 12$ L Niskin bottles.

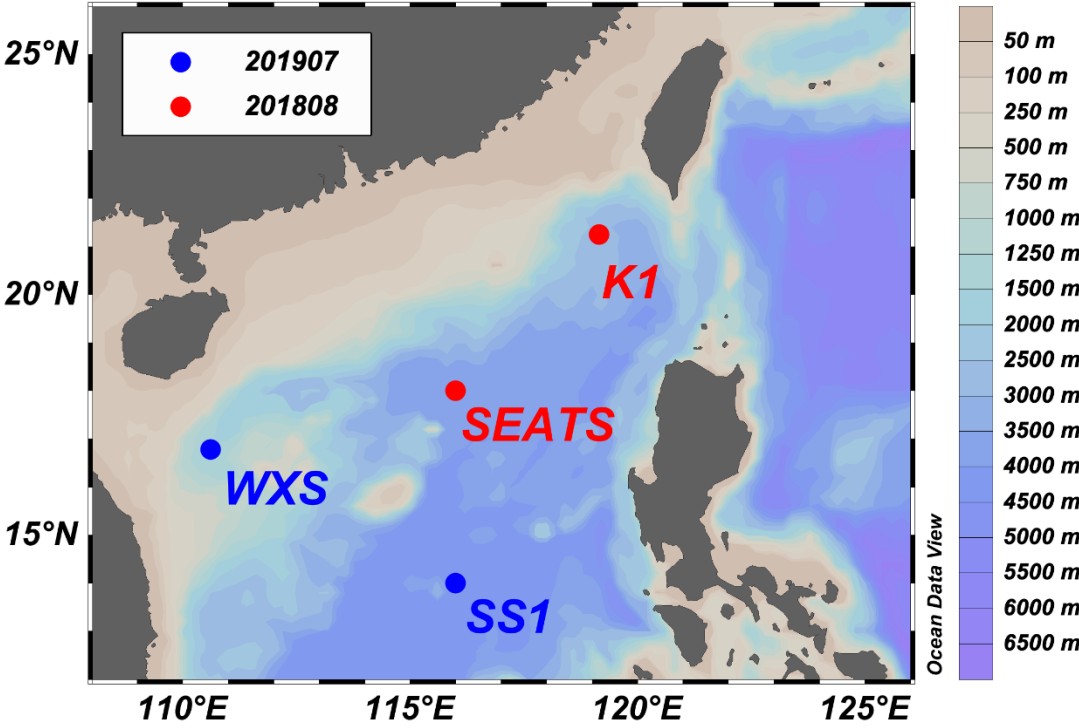


Fig. 1. Station map of two cruises in the South China Sea. Stations visited in Aug 2018 are shown in red and those visited in Jul 2019 are shown in blue. Background colour depicts bathymetry (Schlitzer, Reiner, Ocean Data View, odv.awi.de, 2020).

### 2.2 N$_2$ fixation incubations

N$_2$ fixation incubations, whether of samples collected from photic or aphotic zones, were conducted in the dark and *in situ*

temperature, allowing for direct vertical comparison of rates made under the same conditions. Samples were collected from 4





stations at different depths (K1: 5, 15, 30, 50, 75, 100, 200, 300 and 740 m; SEATS: 5, 15, 30, 50, 75, 100, 200, 300, 705, 1000, 2000 and 3800 m; SS1: 5, 15, 25, 50, 75, 100, 120, 150, 200, 300, 500, 700, 1000 and 4000 m; WXS: 5, 200 and 855 m). Sampling at each station included the oxygen minimum layer (K1: 740 m; SEATS: 705 m; SS1: 700 m; WXS: 855 m), although oxygen concentrations (> 62 μM) were never severely depleted. Water samples from above 100 m were collected

into 1 L polycarbonate bottles (Nalgene) and those below 100 m were collected in acid-cleaned 4 L fluorinated polyethylene (FLPE) bottles. After sampling, waters were kept in the dark with light-blocking black plastic bags until incubations. Fig. 2 provides a schematic of the different incubations conducted during both cruises.

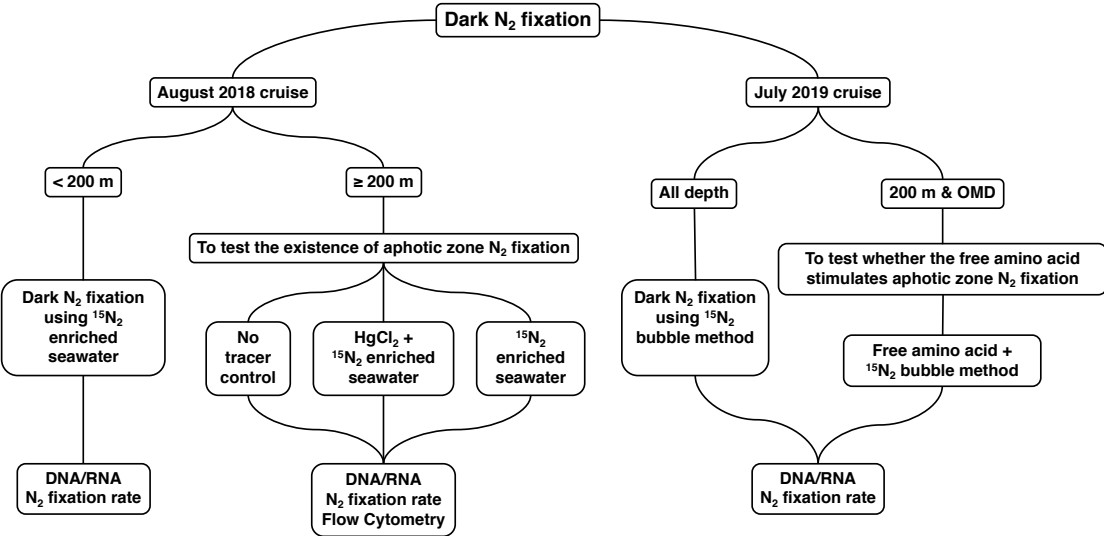

**Fig. 2. Schematic plot of dark $N_2$ fixation experiments conducted during Aug 2018 and Jul 2019 cruises.**

On the August 2018 cruise, $N_2$ fixation rates were measured using the $^{15}N_2$-enriched seawater method (Mohr et al., 2010). However, this method reportedly reduces atom-% enrichment to 2-5 % (White et al., 2020), requiring larger volumes of the enriched seawater for the incubations in environments with low $N_2$ fixation rates. Larger 0.2 μm-filtered enriched seawater amendments dilute PN, decreasing the sensitivity of IRMS measurements. In order to avoid this problem and further enrich seawater with $^{15}N_2$, we applied the $^{15}N_2$ bubble method (Montoya et al., 1996) on the July 2019 cruise. However, this method

can significantly underestimate $N_2$ fixation rates since the $^{15}N_2$ bubble takes about 3 to 15 hours for dissolution (Mohr et al., 2010; Jayakumar et al., 2017; Wannicke et al., 2018; White et al., 2020). Considering that our incubation lasted 24 - 96 hours, this underestimation of rates is largely reduced (Wannicke et al., 2018). Still, our rates should be considered as minimum rates. Details of our experimental design are shown in Table 1. The proportion of $^{15}N_2$ in the incubation bottles was kept < 10 % for $^{15}N_2$-enriched seawater method, and was < 20 % for $^{15}N_2$ bubble method.


**Table 1. Experimental treatments in this study. All incubations were in dark conditions. No tracer and $HgCl_2$ denote tracer-free control and killed control with $HgCl_2$, respectively. PC and FLPE denote polycarbonate and fluorinated polyethylene bottles, respectively.**



| Cruise | Station | Depth (m) | Control Treatment | Method | Replicates | Time Series (hours) | Bottle Type |
|---|---|---|---|---|---|---|---|
| Aug 2018 | K1 | $\leq 100$ | | $^{15}N_2$-Enriched Seawater | 2 | 0, 24 | 1 L PC |
| | K1 | > 100 | | $^{15}N_2$-Enriched Seawater | 4 | 0, 24 | 4 L FLPE |
| | SEATS | $\leq 100$ | | $^{15}N_2$-Enriched Seawater | 2 | 24 | 1 L PC |
| | SEATS | 200, 300, 705, 1000, 2000, 3800 | | $^{15}N_2$-Enriched Seawater | 4 | 0, 48, 96 | 4 L FLPE |
| | SEATS | | $HgCl_2$ | $^{15}N_2$-Enriched Seawater | 2 | 0, 48, 96 | 1 L PC |
| | SEATS | | No tracer | No tracer amendment | 2 | 0, 48, 96 | 1 L PC |
| Jul 2019 | SS1 | $\leq 100$ | | $^{15}N_2$ Gas Bubble | 3 | 0, 24 | 1 L PC |
| | SS1 | 100-200 | | $^{15}N_2$ Gas Bubble | 3 | 0, 24 | 4 L FLPE |
| | SS1 | $\geq 200$ | | $^{15}N_2$ Gas Bubble | 3 | 0, 24, 48 | 4 L FLPE |
| | WXS | 5, 200, 855 | | $^{15}N_2$ Gas Bubble | 3 | 0, 24, 48 | 4 L FLPE |

Samples below 200 m were incubated for two time points, 48 and 96 h for August 2018 cruise and 24 and 48 h for July 2019 cruise. This adjustment was made because we found that isotopic signal of ANF was detectable in 48 h incubation in August 2018 cruise, and so we reduced the incubation time to minimize $^{15}N_2$ leakage as well as community shift and other bottle effects.

During the August 2018 cruise, $^{15}N_2$-enriched seawater was made $\leq$ 24 h prior to the experiments following previous studies
(Lu et al., 2018, 2019; Shiozaki et al., 2015). Due to limited water sample availability, seawater from the surface and 1000 m was collected in the SCS basin area prior to incubation sampling, for 5-100 m and 200-3800 m $^{15}N_2$-enriched seawater, respectively. Surface and 1000 m seawater were filtered (0.22 μm membrane, Millipore) and degassed (STERAPORE 20M1500A membrane, Mitsubishi Rayon Co., Ltd.). Degassed seawater was filled into 2 L Tedlar Bags excluding bubbles, and 20 mL $^{15}N_2$ gas (98.9 %, Cambridge Isotope Laboratories) was injected and mixed manually until all bubbles dissolved.
$^{15}N_2$-enriched seawater of 100 mL or 400 mL were added to the 1 L or 4 L incubation bottles, respectively, resulting in 6 – 10 % or 7-9 % $^{15}N_2$ enrichments assuming complete $^{15}N_2$ dissolution, respectively. Samples were filtered (pre-combusted 25 mm 0.3 μm GF75 (Advantec) under < 400 mbar) immediately after tracer addition to determine the initial (T0) $\delta^{15}N$ of the PN. Incubations were conducted for 24 h for waters $\leq$ 100 m and for 48 and 96 h for waters > 100 m under the dark condition and *in situ* temperature ($\pm$ 6 °C for $\leq$ 100 m samples, and $\pm$ 2.5 °C for all the other samples).
For the July 2019 cruise, incubation bottles were filled with seawater excluding bubbles. T0 samples were filtered immediately after sampling before tracer addition. Atmospheric pressure equilibrated 1 mL and 10 mL of $^{15}N_2$ gas was added through septum caps to 1 L and 4 L incubations, respectively. After tracer injection, bottles were inverted more than 10 times to enhance dissolution. Incubations were conducted in the dark and at the *in situ* temperatures $\pm$ 2.5 °C for 24 h for waters $\leq$ 100 m (except the 5 m sample at WXS station, for which 48 hour incubations were also conducted) and for 24 and 48 h for
waters > 100 m. For time point sampling, incubations were stopped by filtration onto pre-combusted 25 mm 0.3 μm GF75



(Advantec) under < 400 mbar, since GF75 is more efficient in catching < 0.7 μm diazotrophs (Bombar et al., 2018). All sample filters were preserved at – 80 °C until isotopic composition analysis (below).

## 2.3 Control experiments

Two types of control experiments were also conducted: tracer-free incubations and killed controls (+HgCl$_2$) on the August
2018 cruise. Water samples were collected together with the N$_2$ fixation incubation samples from 200 m, 300 m, 705 m, 1000 m, 2000 m, 3800 m at the SEATS station. For tracer-free controls, 1 L polycarbonate bottles were fully filled with seawater without any $^{15}$N$_2$ addition. These experiments were done to determine whether δ$^{15}$N-PN changed over the course of incubations. For the killed controls, after adding 1 mL ca. 0.2 mol L$^{-1}$ HgCl$_2$ solution, 100 mL $^{15}$N$_2$-enriched seawater was added to 1 L polycarbonate bottles to test whether abiotic processes could result in changes in the δ$^{15}$N-PN over the course of
incubations. Control bottles were incubated for 0, 48 and 96 h. Duplicate samples were filtered (pre-combusted 25 mm 0.3 μm GF75 (Advantec) under < 400 mbar) to terminate incubations at each time point. All sample filters were preserved at – 80° C until isotopic composition analyses (below).

## 2.4 Amino acid addition experiments

We hypothesized that the availability of labile organic compounds limits ANF because these compounds occur at low
concentrations in the aphotic zone (Hansell, 2013; Arrieta et al., 2015). To determine whether labile organic matter can stimulate N$_2$ fixation in SCS, N$_2$ fixation incubations with free amino acid additions were conducted in the dark for 200 m and the oxygen minimum layer of SS1 and WXS in the Jul 2019 cruise. Five different volumes of DFAA (a mixture of glutamic acid, glycine, and alanine with the concentration ratio of 1:3:2) were added to the incubation bottles at final concentrations of 0.072, 0.14, 0.72, 3.6, and 7.2 μM N, corresponding 0.2, 0.4, 2, 10, and 20 μM C, representing ca. 0.5, 1, 5,
25, and 50%, respectively, of the total organic carbon (Hansell, 2013; Yamashita and Tanoue, 2003). After mixing the DFAA by inverting the bottles, 10 mL of $^{15}$N$_2$ gas was injected into the bottles and mixed by inversion. All samples were incubated in the dark and under near *in situ* temperature for ca. 2 days. Sampling and sample preservation were conducted as described above.

## 2.5 PN concentration and δ$^{15}$N-PN determination

Filters were freeze-dried and PN subsequently oxidized to nitrate (NO$_3^-$) using the alkaline persulfate digestion method (Knapp et al., 2005). The concentration and isotopic composition of this NO$_3^-$ were then determined via the denitrifier method (Sigman et al., 2001; Casciotti et al., 2002; McIlvin and Casciotti, 2011) using GasBench IRMS (Delta V, Thermo Scientific).

In short, potassium peroxodisulfate (K$_2$S$_2$O$_8$, Merck, 105091) was recrystallized three times for purification, then mixed with
sodium hydroxide (NaOH, Merck, 106498) and ultrapure water, forming peroxodisulfate reagent (POR) with mass ratio of K$_2$S$_2$O$_8$ : NaOH : Milli-Q water = 6 : 6 : 100. Then, POR blanks were tested before each digestion experiment. POR blanks





were autoclaved under 105 °C for 70 min, and their pH was adjusted to ~ 6 by using 6 mol L$^{-1}$ hydrochloric acid (HCl, Merck, 100317), after which NO$_3^-$ concentrations were determined using the chemiluminescence method (Braman and Hendrix, 1989). POR was only used when its blank was below 2 μM N (Fig. S1 *a*). A volume of 0.5 mL of POR was added
to each sample, which was digested in the same procedure as POR blanks.

The N added by the POR was below 1 nmol, < 4 % of the total sample N. Meanwhile, the isotopic composition (δ$^{15}$N) of blanks in each batch were stable (standard deviation (SD) < 2.5 ‰). Likewise, the filter blank in our experiment was 11.86 ± 2.73 nmol N (Fig. S1 *b*), which accounted for 14.0 ± 4.5 % and 4.6 ± 3.3 % of the N content of the samples in the August 2018 and July 2019 cruises, respectively. Together, these low concentration and stable isotopic composition of the blank of
POR and filter demonstrated their limited impact on sample measurement, which could be further reduced after calibration.

Recovery (102 ± 3.6 %) and reproducibility (SD = 0.3 ‰) of this digestion method were determined using known amounts of urea and EDTA (Fig. S1 *b*). δ$^{15}$N was -1.5 ± 0.3 ‰ for urea and -0.4 ± 0.1 ‰ for EDTA. These were not significantly different from δ$^{15}$N values acquired via EA-IRMS which were -2.0 ± 0.03 ‰ and -0.4 ± 0.05 ‰ for urea and EDTA, respectively (Fig. S1 *b*). Both recovery rates and δ$^{15}$N were stable up to 2000 nmol of urea and EDTA (Fig. S1 *b*), suggesting
negligible isotope fractionation effect during the digestion process. Together, the low blank and stable recovery of samples during the digestion experiment enable a precise measurement of the concentration and isotopic composition down to 20 nmol PN in our study.

The minimum absolute amount of N in PN required for persulfate-denitrifier method is 20 nmol, which is the amount needed for IRMS to get stable δ$^{15}$N-N$_2$O values (Fig. S1 *d*). However, in order to minimize biases caused by blanks of GF75 and
POR, the recommended absolute amount of PN for this method is ≥ 100 nmol.

**2.6 N$_2$ fixation rate calculation**

N$_2$ fixation rates were derived according to Montoya et al. (1996):

$$SNFR = \frac{1}{\Delta T} \times \frac{A_{PN_T} - A_{PN_0}}{A_{N_2} - A_{PN_0}} \qquad (1)$$

$$NFR = SNFR \times \frac{[PN]_T + [PN]_0}{2} \qquad (2)$$

where $SNFR$ is the specific N$_2$ fixation rate in d$^{-1}$, $NFR$ is N$_2$ fixation rate in nmol N L$^{-1}$ d$^{-1}$. $\Delta T$ represents incubation time, and $A_{PN_0}$ and $A_{PN_T}$ represent isotope ratio ($^{15}$N/($^{14}$N+$^{15}$N)) of PN at incubation start and end, respectively. $[PN]_T$ and $[PN]_0$ represent PN concentrations at incubation start and end, respectively. $A_{N_2}$ is the isotope ratio of substrate N$_2$ during the incubation.

$A_{N_2}$ was measured using Membrane Inlet Mass Spectrometer (MIMS) in N$_2$ fixation incubations at 500 m, 700 m and 4000 m in SS1 and 5 m and 200 m in WXS from the July 2019 cruise (Kana et al., 1994). $A_{N_2}$ samples were collected with gas tight syringes and tubes from incubation bottles at the final time point and transferred to Exetainer$^{TM}$ vials excluding





bubbles. Dissolved $^{15}N_2$ standards for MIMS were also prepared by adding different amount of $^{15}N_2$ (Cambridge Isotope Laboratories) into air-equilibrated Milli-Q water, followed by 72 hours equilibration period, and measured along with $A_{N_2}$

samples (White et al., 2020). The $A_{N_2}$ was $15 \pm 2$ % (average $\pm$ error) for MIMS samples below 200 m, thus we set $A_{N_2}$ to be $15 \pm 2$ % for $N_2$ fixation samples with the same amount of $^{15}N_2$ addition (i.e. 10 mL in 4 L bottles) without $A_{N_2}$ measurements in July 2019 cruise. Samples < 200 m had large temperature variation, so we used the ratio of measured and theoretical $A_{N_2}$ assuming complete $^{15}N_2$ dissolution to calibrate $A_{N_2}$. The ratio was $46.7 \pm 8.9$ % at 5 m in WXS. Thus, $A_{N_2}$ for samples < 200 m was set to be $46.7 \pm 8.9$ % of theoretical $A_{N_2}$. Unfortunately, we weren't able to sample $A_{N_2}$ in

August 2018 cruise. Thus, $A_{N_2}$ were calculated with the amount of added tracer, salinity and temperature (Hamme and Emerson, 2004) assuming complete dissolution.

Depth integrated $N_2$ fixation rates were calculated by multiplying the average of two adjacent rates by their depth interval. The shallowest measurement is 5 m, within the upper mixed layer. Thus, 0-5 m $N_2$ fixation rate integration was calculated using rates at 5 m assuming constant rates. Since ANF rates below 1000 m were undetectable, and the WXS station only had

three sampling depths, the depth integrated $N_2$ fixation rates were only calculated for upper 1000 m of K1, SEATS and SS1.

## 2.7 Analytical considerations for dark $N_2$ fixation rates

Reliable $N_2$ fixation rates come from both sufficient PN mass for isotopic determination and the isotopic change of PN ($\Delta\delta^{15}N$-PN) exceeding the minimum acceptable change for $N_2$ fixation. As mentioned above, 20 nmol PN is the lowest determination amount for stable isotopic composition using the persulfate-denitrifier method, and all the PN samples were

above this limit.

The minimum acceptable change for $\delta^{15}N$-PN were set to be average + 3 SD of $\delta^{15}N$-PN in T0, which is calculated for each sampling depth. This criterium accounts for both the analytical detection limit and natural $\delta^{15}N$-PN variability. Given the stable isotopic analyses mentioned above, we consider detectable biological $N_2$ fixation occurs when at least one data point after incubations surpasses the minimum acceptable $\delta^{15}N$-PN change. As a result, only $N_2$ fixation rates in 1000 m, 2000 m

and 3800 m at SEATS, 15 m, 100 m, 500 m, 1000 m and 4000 m at SS1 were below this detection limit (8 out of 39 layers). Also, among those detectable rates, 30 out of 31 layers had at least 2 data points after incubations surpased the minimum acceptable $\delta^{15}N$-PN change. $N_2$ fixation rate at 705 m at SEATS was also considered as below detection limit since control experiments after incubation had $\delta^{15}N$-PN above the minimum acceptable change.

We also calculated the detection limit according to propagation of experimental uncertainty associated with each measured

parameter following White et al. (2020) and Montoya et al. (1996). As a result, only 5-100 m at SEATS, 5-75 m at K1, 5 m, 25 m, 75 m at SS1 and 5 m at WXS were above this detection limit. However, we think the $\delta^{15}N$-PN after incubation surpassing the minimum acceptable change is the direct and solid evidence for the occurrence of biological $N_2$ fixation. When error propagation is taken into account, the isotopic signal for biological $N_2$ fixation could be merged. Thus, in this




study we applied the minimum acceptable $\delta^{15}$N-PN change as the criteria for the occurrence of biological N$_2$ fixation,
although these rates have large uncertainties after error propagation.

## 2.8 Cell abundance and mean diameter

Cell abundance and mean diameter data were obtained using a FACSAria flow cytometer following Jiao et al. (2005). 1.98
mL of seawater were collected just before the PN sampling at each time point, and fixed with 20 μL 50% glutaraldehyde.
After mixing, samples were incubated in the dark for 15 min before being flash-frozen in liquid N$_2$. Samples were then
stored at -80° C until further analysis.

Before analysis, samples were thawed and stained with SYBR Green I (Invitrogen). Beads (1 μm) were added as an internal
standard. Green fluorescence (FITC-A) and side scatter (SSC-A) were used to count cells using FlowJo software. Mean cell
diameter was calculated from side scatter (SSC-A) data using a linear regression model of heterotrophs (Calvo-Díaz and
Morán, 2006).

## 2.9 Data compilation

We compiled data from 17 published ANF rate studies (Table 2) including location, depth range, volumetric and depth-
integrated N$_2$ fixation rates, aphotic zone N$_2$ fixation contribution, and method used for rate determination. These studies
covered a wide geographic range, from coastal and marginal seas to open oceans. Only N$_2$ fixation data below the euphotic
zone were included. For studies that did not report euphotic zone depth, only data below 200 m were compiled.

**Table 2. Compilation of N$_2$ fixation rate studies conducted below the euphotic zone using $^{15}$N$_2$ methods. For studies without explicit
depth of euphotic zone, only data ≥ 200 m were included. BDL stands for below detection limit.**

| Location | Depth (m) | N$_2$ fixation rate (nmol N L$^{-1}$d$^{-1}$) | Aphotic zone contribution to total % | Aphotic zone integrated NFR (μmol N m$^{-2}$ d$^{-1}$) | Method | Reference |
|---|---|---|---|---|---|---|
| *Hypoxic waters* | | | | | | |
| Southern California Bight | 500, 885 | 0.7 | ca. 30 % (below DCM to 855 m) | 55 | $^{15}$N$_2$ bubble | Hamersley et al., 2011 |
| Eastern Tropical South Pacific | OMZ core (deepest 400 m) | 1.27 ± 1.2 | ca. 90 % (deepest level of 1 μmol L$^{-1}$ O$_2$ to 400 m) | 574 ± 294 | $^{15}$N$_2$ bubble | Fernandez et al., 2011 |
| Baltic Sea | 200 | 0.44 ± 0.26 | $^a$6 % (Suboxic and anoxic area) | Not reported | $^{15}$N$_2$ bubble | Farnelid et al., 2013 |
| Eastern Tropical South Pacific | 200-2000 | BDL-0.6 | 87-90 % (below the euphotic zone to 2000 m) | 119-501 | $^{15}$N$_2$ bubble | Bonnet et al., 2013 |
| Eastern Tropical South Pacific | 200 | 0.37 | Not reported | Not reported | $^{15}$N$_2$ bubble | Dekaezemacker et al., 2013 |
| Peruvian OMZ | 200 | 0.4 | Not reported | Not reported | $^{15}$N$_2$ bubble | Loescher et al., 2014 |
| Eastern Tropical South Pacific | 200-500 | BDL-4.39 | Not reported | 150.6-628.7 (0-500 m) | $^{15}$N$_2$ enriched seawater | Löscher et al., 2016 |
| Pacific Northwest coastal upwelling system | 600 | BDL | Not reported | Not reported | $^{15}$N$_2$ enriched seawater | Gradoville et al., 2017 |
| Eastern Tropical South Pacific | 200-350 | BDL | Not reported | Not reported | Modified $^{15}$N$_2$ bubble | Chang et al., 2019 |



| | | | | | | |
|---|---|---|---|---|---|---|
| Eastern Tropical North Pacific | 110-3001 | BDL-35.9 | Not reported | Not reported | Modified [15]N$_2$ bubble | Selden et al., 2019 |
| *Without hypoxia* | | | | | | |
| Levantine Basin | 250-500 | 0.01-0.24 | 37-75 % (0.1 % PAR to 500 m) | Not reported | [15]N$_2$ bubble | Rahav et al., 2013 |
| Gulf of Aqaba | 150-720 | 0.02-0.38 | 56 % (0.1 % light to 720 m) | Not reported | [15]N$_2$ bubble | Rahav et al., 2013 |
| Gulf of Aqaba | 200 | 0.2-0.3 | Not reported | Not reported | [15]N$_2$ bubble | Rahav et al., 2015 |
| Gulf of Mexico | 330-538 | [b]1.06 ± 0.24 * 10[-5] h[-1] | Not reported | Not reported | [15]N$_2$ bubble | Weber, 2015 |
| Solomon Seas | 200-1000 | BDL-0.35 | [c]25 % (200-1000 m) | Not reported | [15]N$_2$ enriched seawater | Benavides et al., 2015 |
| Bismarck Seas | 200-1000 | BDL-0.89 | [c]25 % (200-1000 m) | Not reported | [15]N$_2$ enriched seawater | Benavides et al., 2015 |
| Mediterranean Sea | 200-2000 | BDL-0.07 | 48-100 % (below 0.01 % PAR to 2000 m) | 17.83-91.06 | [15]N$_2$ bubble | Benavides et al., 2016 |
| North Pacific Subtropical Gyre | 200 | BDL | Not reported | Not reported | [15]N$_2$ enriched seawater | Gradoville et al., 2017 |
| Western Tropical South Pacific | 200-800 | 0.05-0.68 | [d]ca. 6-88 % (200-800 m) | Not reported | [15]N$_2$ bubble | Benavides et al., 2018 |
| Western North Atlantic Coastal waters | 19.8-40.5 | 1.77-9.12 | Not reported | Not reported | [15]N$_2$ bubble | Mulholland et al., 2019 |
| South China Sea | 200-4000 | BDL-0.21 | [e]39 ± 32 % (200-1000 m) | 36 ± 26 | [15]N$_2$ enriched seawater/[15]N$_2$ bubble | This Study |

[a]Contribution of annual surface water NF in Baltic Sea
[b]Shown in SNFR
[c]Of 5-70 m integration
[d]Of photic zone integration
[e]Of 0-1000 m dark integration


In order to resolve the relationship between observed detectable ANF, SNFR and PN, we only compiled values collected from depths below 200 m. We also discarded studies reporting too few data. For depths without SNFR data, SNFR were derived by dividing ANF by PN. In this study, 13 out of 18 ANF depths were compiled (Southern California Bight (SCB):

14 of 21 ANF depths (Hamersley et al., 2011), Eastern Tropical South Pacific (ETSP): 31 of 39 (Bonnet et al., 2013), 2 of 99 (Löscher et al., 2016), Mediterranean Sea: 10 of 10 (Benavides et al., 2016), Western Tropical South Pacific (WTSP): 59 of 59 (Benavides et al., 2018a), and ETNP: 8 of 51 ANF depths (Selden et al., 2019)).

### 3 Results

### 3.1 N$_2$ fixation rates

The isotopic signal of three treatments (tracer-free control, abiotic control, and ANF) in the August 2018 cruise are shown in Fig. 3. The changes of $\delta^{15}$N-PN for each replicate in tracer-free and abiotic controls were below the minimum acceptable change (average + 3 SD of $\delta^{15}$N-PN at T0), except 705 m at SEATS station, suggesting that processes other than N$_2$ fixation yielded negligible isotopic change of PN.





**Fig. 3. Isotopic composition of control experiments conducted at SEATS station on Aug 2018 cruise. The grey line stands for the lowest $\delta^{15}N$-PN value needed for detectable $N_2$ fixation rates (average + 3 SD of $\delta^{15}N$-PN at T0) after incubation.**



Most $N_2$ fixation experiments at 200-1000 m (11 out of 15 depths) yielded detectable rates with at least 2 data points above the minimum acceptable $\delta^{15}$N-PN change after 1 to 4 days' incubations, indicating the occurrence of ANF in SCS (Fig. 3 and Fig. 4). Below 1000 m, neither detectable ANF rates observed, nor $\delta^{15}$N-PN after incubation differed from tracer-free and

abiotic controls, indicating the absence of ANF (Fig. 3 and Fig. 4). The highest rates in the water column occurred in the depth range of 25 to 75 m ($0.27 \pm 0.14$-$1.28 \pm 0.85$ nmol N L$^{-1}$ d$^{-1}$). Nevertheless, ANF was detected at all stations at low rates (< 0.2 nmol N L$^{-1}$ d$^{-1}$). In general, ANF rates and SNFR exhibited a decreasing trend with depth except station K1 and SS1, which had peaks at the OMD depth (Fig. 5).









**Fig. 4. Isotopic composition of ANF incubation experiments in 200 m *(a)*, 300 m *(b)*, OMD (SEATS: 705 m; K1: 740 m; SS1: 700 m; WXS: 855 m) *(c)*, 1000 m *(d)*, bottom (SEATS: 3800 m; SS1: 4000 m) *(e)* depth conducted in both Aug 2018 and Jul 2019 cruises. Each symbol represents one replicate.**

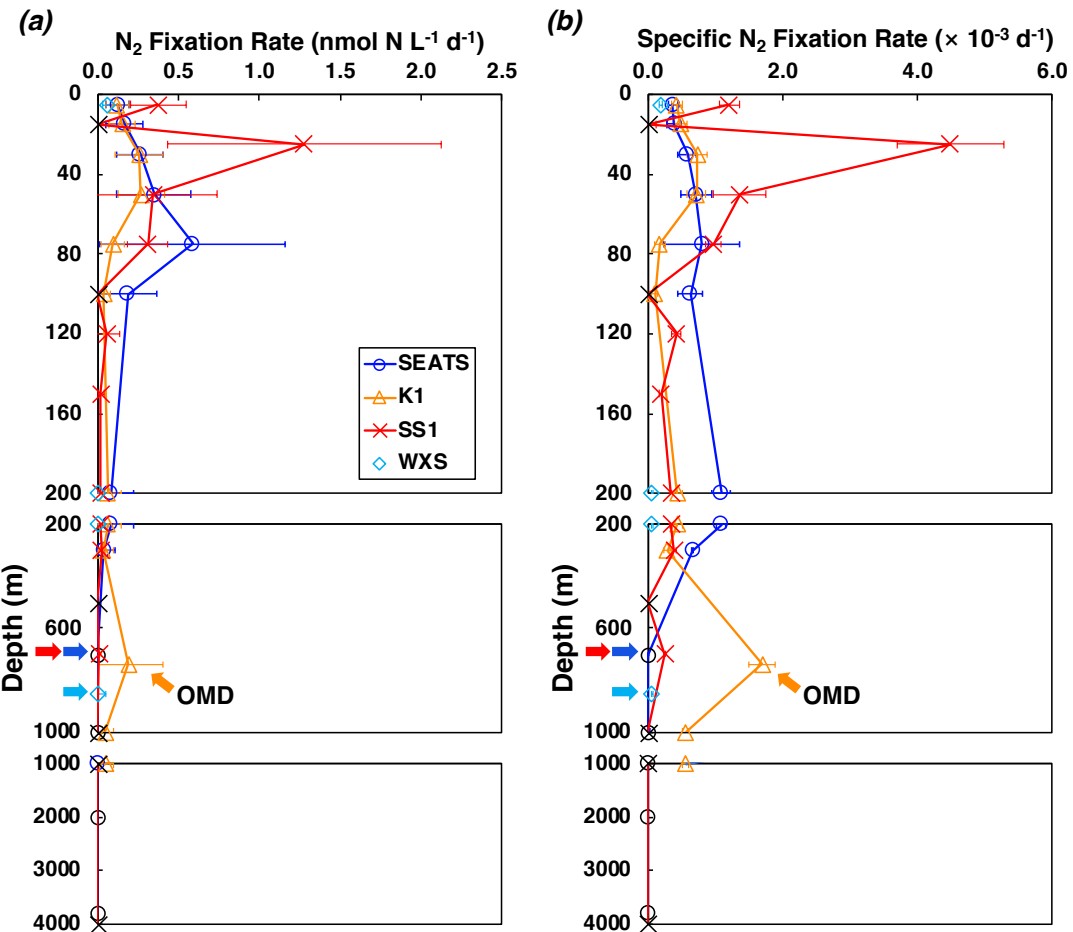

**Fig. 5. Vertical profiles of N₂ fixation rates *(a)* and specific N2 fixation rates *(b)* under dark conditions in station SEATS (circle),**
**K1 (triangle), SS1 (cross), WXS (diamond). Arrows represent the depth of OMD (> 2 mg L⁻¹). Black symbols represent rates below detection limit.**

The 0-1000 m integrated rates were $108 \pm 77$, $61 \pm 27$, and $51 \pm 19$ µmol N m$^{-2}$ d$^{-1}$ for stations K1, SEATS and SS1, respectively (Fig. 6), and the 200 to 1000 m depth interval accounted for $39 \pm 32$ % of the integrated rates within the upper 1000 m of water column.





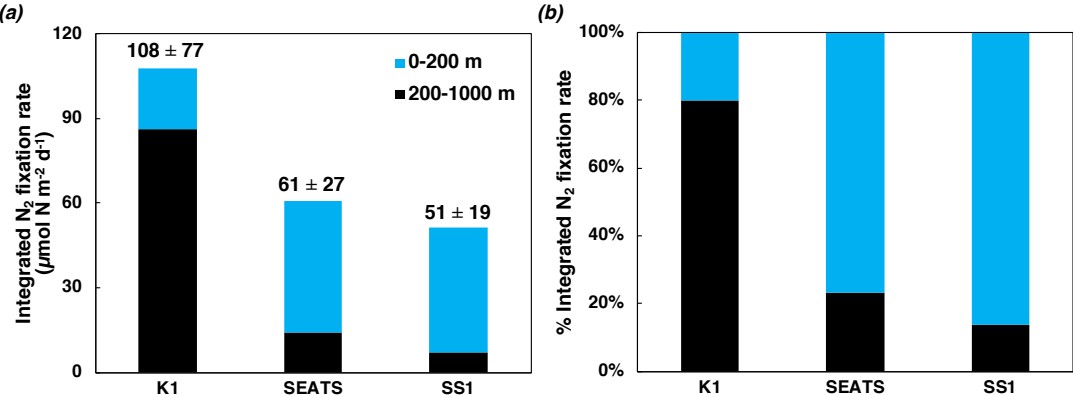


**Fig. 6. Integrated dark N₂ fixation rates at station K1, SEATS and SS1 *(a)*. WXS station is not included since only 3 depths were measured. Numbers above bars in *(a)* represent integrated rates from 0-1000 m. Blue and black colours indicate the 0-200 m and 200-1000 m depth ranges, respectively. *(b)* percent (%) contribution of 0-200 m and 200-1000 m to 0-1000 m integrated rates.**

### 3.2 Most amino acid addition experiments observed no ANF enhancement

DFAA addition experiments were conducted with five DFAA final concentrations at 200 m and OMD layer (700 m at SS1 and 855 m at WXS) at SS1 and WXS stations. Only 7 out of 20 DFAA addition experiments yielded detectable N₂ fixation rates in this study (Fig. 7 and Fig. 8), and only 3 of them yielded enhanced N₂ fixation rates compared to no DFAA addition controls (Fig. 8). These enhancements only occurred at 200 m, with medium DFAA additions of 0.4-10 µmol C L⁻¹. The lowest DFAA addition of 0.7 µmol N L⁻¹ (0.2 µmol C L⁻¹) at all depths and all DFAA addition treatments at 700 m of SS1

yielded no detectable N₂ fixation rates (Fig. 8). At 855 m of WXS, 1.4 µmol N L⁻¹ (0.4 µmol C L⁻¹) and 7 µmol N L⁻¹ (2 µmol C L⁻¹) DFAA addition yielded detectable yet lower rates compared to the control.





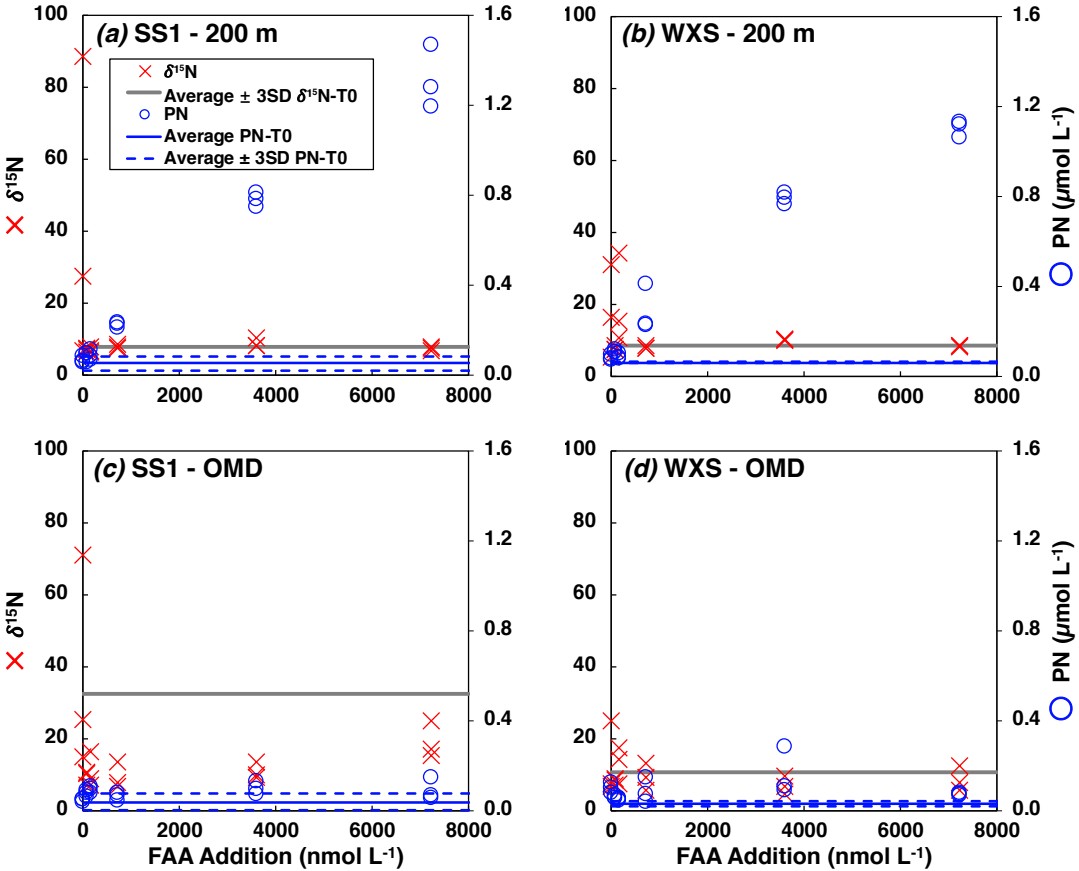

**Fig. 7.** δ15N-PN (red crosses) and PN concentrations (blue dots) after DFAA addition incubations in *(a)* 200 m of SS1, *(b)* OMD (700 m) of SS1, *(c)* 200 m of WXS, and *(d)* OMD (855 m) of WXS. Gray solid lines represent minimum acceptable change of δ15N-PN for N₂ fixation. Blue solid and dotted lines represent average PN concentrations and average ± 3 SD in T0 time point (without ¹⁵N₂ nor DFAA addition), respectively.



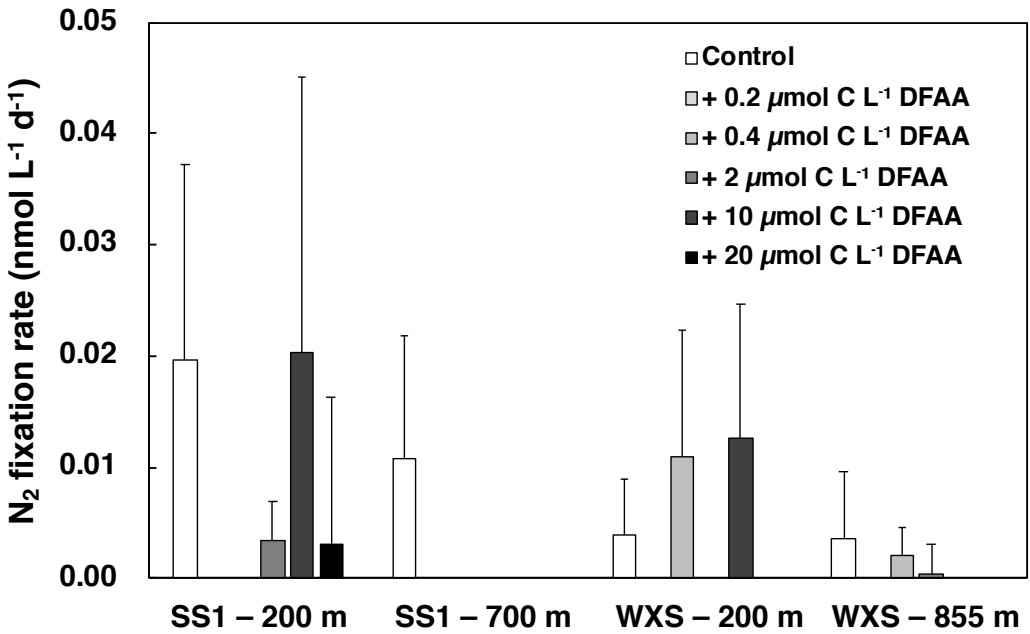

**Fig. 8. N₂ fixation rates with DFAA additions at 200 m and OMD (700 m for SS1 and 855 m for WXS) at SS1 and WXS. All rates are term averaged rates and error bars are derived via error propagation. Controls represent N₂ fixation rates without DFAA additions.**

Interestingly, the addition of DFAA increased the PN concentration in most of our incubations (Fig. 7). The linear increases of PN at the 200 m depth of SS1 and WXS indicated active N assimilation into the particulate phase.

## 4 Discussion

### 4.1 Methodological attempts to determine low ANF rates

Aphotic zone waters are characterized by low PN concentrations and low ANF rates, making the detection of biological $N_2$ fixation more challenging than in euphotic waters. As IRMS linearity decreases with decreasing mass, both a sufficient amount of PN for isotopic determination and $\Delta\delta^{15}N$-PN above minimum detectable change are essential for detectable rates (Moisander et al., 2017; White et al., 2020).

In this study, we employed persulfate-denitrifier method to determine $\delta^{15}N$-PN and PN mass, reducing the minimum PN mass needed to determine isotopic enrichment from 10 µg for the EA-IRMS method (White et al., 2020) to 1.4 µg (100 nmol N). Thus, incubation volumes can be reduced, reducing efforts for large volume collection and filtration. For example, for environments with 0.05 µmol N L⁻¹ PN, such as in the mesopelagic SCS, ca. 14 L seawater is needed to reach recommended 10 µg N for EA-IRMS analysis of $\delta^{15}N$-PN, while only 2 L seawater is needed for recommended 100 nmol in this method. By reducing the water budget, our approach reduces ship-time and labor costs, allowing for more increased sampling





coverage. Therefore, this method would be of great use also in other low N₂ fixation systems such as estuarine environments

or ultraligotrophic waters, or even polar waters, expanding the geographical limits of current N₂ fixation measurements.

There are two ways to enhance $\Delta\delta^{15}$N-PN. One is to increase incubation time, and the other is to increase $A_{N_2}$. However,

increasing incubation time may risk enhanced bottle effects, such as community shift and seawater chemical change, as well

as the movement of the fixed $^{15}$N to other pools such as excretion or cell lysis. Thus, increasing $A_{N_2}$ is preferable. Both the

$^{15}$N₂ bubble and the $^{15}$N₂ enriched seawater methods were used in this study. Their pros and cons were well reviewed by

White et al. (2020). Introducing $^{15}$N₂ enriched seawater to achieve $A_{N_2}$ >10 % risks nutrient and/or trace metal contamination

(White et al., 2020). However, $A_{N_2}$ > 10 % can be easily achieved by using the $^{15}$N₂ bubble method without this risk.

Nevertheless, incomplete $^{15}$N₂ gas dissolution (Jayakumar et al., 2017; Mohr et al., 2010; White et al., 2020; Wannicke et al.,

2018) with the $^{15}$N₂ bubble method (Montoya et al., 1996) may result in underestimation of rates. Here, we used this method

to maximize isotopic signals to examine the occurrence of ANF. Thus, following studies are recommended to use the more

precise modified $^{15}$N₂ bubble method (Klawonn et al., 2015).

We also set tracer-free and abiotic control experiments to evaluate influence of processes other than biological N₂ fixation.

$\delta^{15}$N-PN replicates after 2-4 days' incubations in controls yielded values below minimum acceptable change (average + 3 SD

of $\delta^{15}$N-PN in T0), except 705 m at SEATS. This suggests that the effects of processes other than biological N₂ fixation were

negligible in ANF measurements, further confirming the reliability of detected ANF rates.

**4.2 Factors affecting the distribution and heterogeneity of dark N₂ fixation in SCS**

Despite the careful quality control measures discussed above, variability was observed among replicate measurements of PN

concentration and $\delta^{15}$N-PN below 200 m (Fig. 3, 4 and S2). The divergence of $\delta^{15}$N-PN replicates in T0 time points was

similar in the August 2018 and July 2019 cruises (Fig. 4), despite methodological differences (see above). This suggests that

the water collected from the same depth had different PN concentrations and $\delta^{15}$N-PN. This might be due to the

heterogeneity of suspended/sinking particles. The larger variances in the sample after the incubation compared to T0 time

points further indicated the heterogeneity of ANF in deep waters (Fig. 3, 4 and S2). How to exactly quantify ANF rates

accounting for such heterogeneity is also an important task for future ANF studies.

ANF heterogeneity is observed not only in small scales (within one sampling depth) but larger spatial scales (across depths

and stations). The highest dark N₂ fixation rates in our study were observed in the euphotic zone (25-75 m), suggesting that

the euphotic zone is an active site for N₂ fixation by either UCYN-B, UCYN-C or proteobacterial diazotrophs, which are

known to upregulate nitrogenase structural genes (*nifH*) in the dark (Church et al., 2005; Taniuchi et al., 2012). Their *nifH*

genes and transcripts are also frequently observed in SCS and other regions of the western Pacific Ocean (Chen et al., 2019;

Liu et al., 2020; Lu et al., 2019; Taniuchi et al., 2012; Zhang et al., 2011). Below the rate maxima, N₂ fixation rates tended to

decrease rapidly with depth (Fig. 5 *b*); however, second peaks were observed in the OMD layer at K1 and SS1, supporting

that the energetic cost of N₂ fixation might be reduced in low oxygen environments (Großkopf and LaRoche, 2012; Riemann





et al., 2010). Thus, the OMD layer could represent a niche for ANF. In our study, ANF rates were below the detection limit under 1000 m, while detectable abyssopelagic and bathypelagic $N_2$ fixation was reported in the Mediterranean Sea (Benavides et al., 2016).

Horizontally, dark $N_2$ fixation patterns in the 0 to 200 m depth interval were similar among stations, with the highest rate observed at SS1. Below the euphotic zone, $N_2$ fixation in the OMD at K1 was an order of magnitude higher than at other stations (Fig. 5 *b*). Accordingly, K1 had the highest 200-1000 m ANF integration rate, followed by SEATS and SS1 (Fig. 6 *a*). As labile organic matter is thought to be the energy source for ANF (Bombar et al., 2016), we speculate that the sinking particle flux may be a key determinant of ANF, i.e., high sinking particle flux may result in high ANF rates, as other studies

have pointed out (Bonnet et al., 2013; Fernandez et al., 2011; Riemann et al., 2010). This is supported by the observation of higher sinking particle fluxes in the northeast SCS (adjacent to K1) than in the central SCS (close to SEATS and SS1) (Yang et al., 2017). On the other hand, the higher total cell abundance, larger mean cell diameter and higher PN concentrations at K1 between 200-1000 m depth further suggest higher microbial activity possibly due to high sinking particle flux (Fig. 9). High integrated ANF rates were also observed in the eastern tropical Pacific (Bonnet et al., 2013; Fernandez et al., 2011;

Löscher et al., 2016), which is also characterized by high sinking particle flux. Together, these lines of evidence suggest that sinking particle flux might be a factor supporting ANF in the aphotic ocean. Other processes such as water mass sinking may also fuel ANF by transporting surface organic matter to deep waters (Benavides et al., 2016).

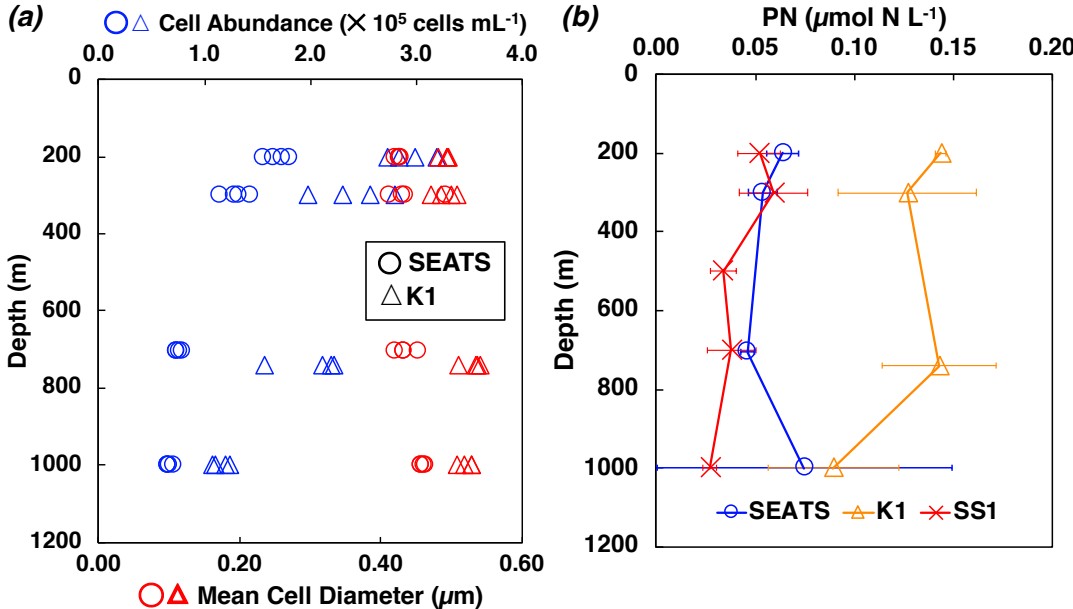

**Fig. 9.** *(a)* Cell abundance (blue) and mean cell diameter (red) of 4 replicates at T0 time points at 200-1000 m of SEATS (open
circles) and K1 (cross symbols) stations derived from flow cytometry analysis. *(b)* PN concentration profiles from 200 m to 1000 m
in SEATS (red), K1 (cyan) and SS1 (blue) stations.





### 4.3 Potential problems and insights of DFAA addition

Stimulation of ANF by DFAA enrichment has been reported in several studies with DFAA enrichment of 0.5-40 μmol C L$^{-1}$ (Table S2) (Benavides et al., 2015; Bonnet et al., 2013; Rahav et al., 2013, 2015; Selden et al., 2019). We conducted DFAA
addition of five concentrations extending two orders of magnitude (0.2-20 μmol C L$^{-1}$, covering DFAA addition in most studies) at 4 depths. However, only 3 out of 7 detectable rates in 20 treatments yielded ANF enhancements while others had lower rates than controls (Fig. 8). This could be because DFAA additions were not sufficient to stimulate a detectable enhancement, or ANF was limited by other factors in the SCS, or the organisms present preferred DFAA assimilation to N$_2$ fixation as a means of getting N. On the other hand, most DFAA addition treatments yielded PN enhancements after
incubation (Fig. 7). Such increases of PN could be due to either amino acid assimilation or dissimilatory amino acid utilization and subsequent nitrate uptake. Considering that we added non-$^{15}$N enriched amino acid, the PN increment could dilute δ$^{15}$N-PN signal from N$_2$ fixation, which is also one possible reason for under-detectable ANF rates with DFAA additions.

Other studies of DFAA additions have resulted in different effects, from a decrease in ANF to a ca. 7-fold increase (Table
S2) (Benavides et al., 2015; Bonnet et al., 2013; Rahav et al., 2013, 2015; Selden et al., 2019). This inconsistency may result from population-specific substrate preferences, non-cyanobacterial diazotrophic community composition (Bentzon-Tilia et al., 2015b), variability in energy or carbon limitation (Selden et al., 2019), metabolic rate and affinity difference caused by different temperature (Nedwell, 1999; Price and Sowers, 2004), or spatial variance of ANF limiting factors. Coupled molecular and advanced stable isotope labeling techniques such as nanoscale secondary ion mass spectrometry (nano-SIMS)
or stable isotope probing are needed to unveil the impact of DFAA on ANF.

### 4.4 Insights revealed by global ANF compilation

To further explore the distribution, potential control and the importance of ANF, we compiled published ANF rates across large environmental gradients (Table 2). Interestingly, studies without hypoxia listed in Table 2, including ours, yielded ANF rates of the same magnitude, regardless of different incubation ($^{15}$N$_2$ bubble method from Montoya et al. (1996), or $^{15}$N$_2$
enriched seawater from Mohr et al. (2010)) and δ$^{15}$N-PN analyses (EA-IRMS or persulfate-denitrifier method). This not only supports the practicality of our method using $^{15}$N$_2$ bubble and persulfate-denitrifier method, but also suggests that the spatial and temporal variability of ANF is larger than the variability caused by different methods (Chang et al., 2019).

In general, the higher ANF rates were reported in studies conducted in hypoxic regions (from below detection limit to 35.9 nmol N L$^{-1}$ d$^{-1}$, 3 of 10 studies reported rates > 1 nmol N L$^{-1}$ d$^{-1}$, Table 2) compared with oxic region studies where 7 of 9
studies reported rates < 1 nmol N L$^{-1}$ d$^{-1}$ with the maximum rate of 9.12 nmol N L$^{-1}$ d$^{-1}$ (Table 2). However, it is worth noting that the reported N$_2$ fixation rates in hypoxic regions were patchy (Selden et al., 2019), indicating that oxygen concentration is not the only determinant of ANF. Hypoxic regions often underlie highly productive surface waters (Lam and Kuypers, 2011), and high particle export from surface waters provides labile organic matter to the hypoxic zones which could provide





an energy source for non-cyanobacterial diazotrophs. Further, some non-cyanobacterial diazotrophs can degrade refractory

organic matter such as aromatic hydrocarbons, providing an additional energy source (Bentzon-Tilia et al., 2015a).
Diazotrophs inhabiting the aphotic zone need to balance the energetic cost of $N_2$ fixation against their energy sources.
However, the quantitative effects of sinking particles and oxygen on ANF in hypoxic regions requires further elucidation.
PN and SNFR are two important components in derivation of ANF rates. PN is a conventional parameter in
biogeochemistry, with much easier determination than ANF. SNFR is the specific rate, revealing the $N_2$ fixation capacity per

PN unit. We looked into detectable PN, SNFR and ANF data from the compiled studies, and divided them into four groups
of oceanic regions: 1) ETNP; 2) SCB and ETSP; 3) SCS and Mediterranean Sea, 4) WTSP (Fig. 10 *a*, *c*). The ETNP data
were self-grouped since one station reported high ANF rates near the inner Revillagigedo Islands (Selden et al., 2019). The
SCB and ETSP data were characterized by high PN and low SNFR while the SCS and Mediterranean Sea were characterized
by low PN and low SNFR. The WTSP was distinguished by high SNFR.

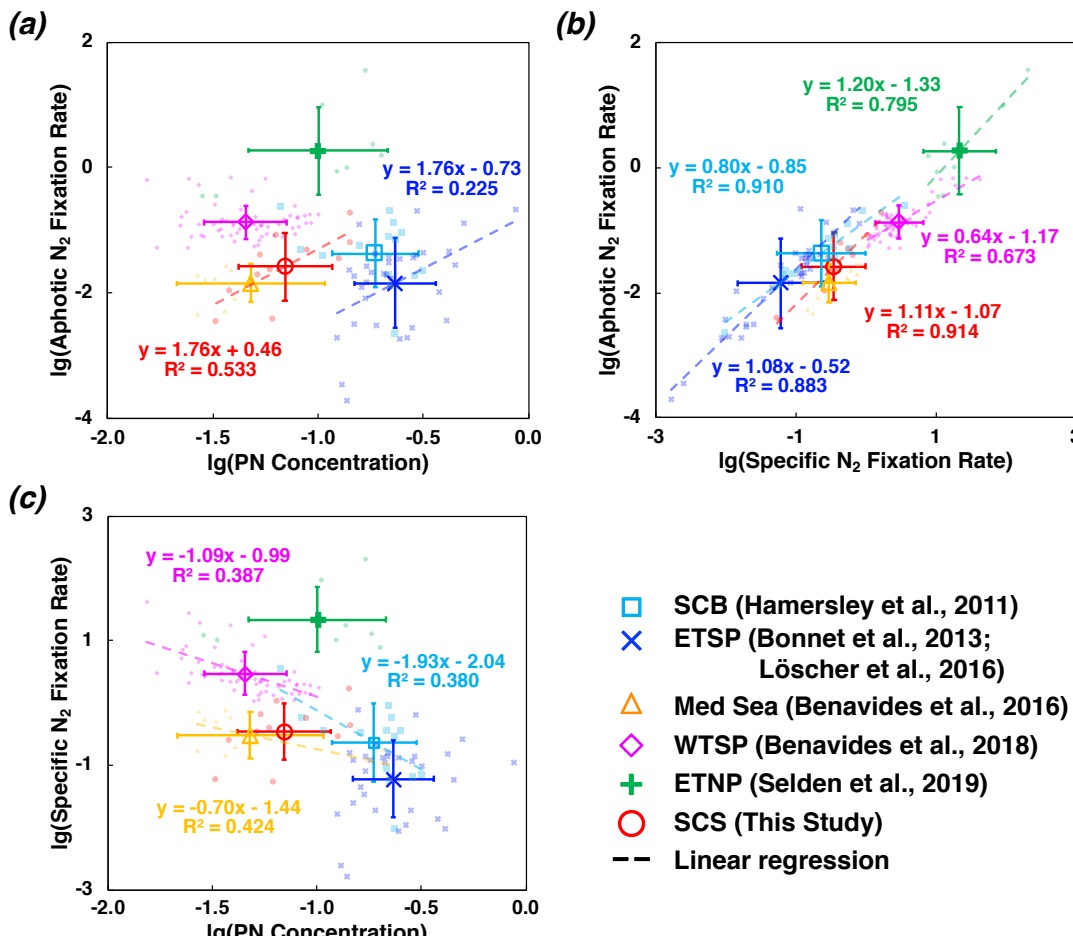


**Figure 10. After taking the logarithm, correlation between ANF and PN concentration *(a)*, ANF and SNFR *(b)*, SNFR and PN concentration *(c)*. Different symbols and colours depict ANF studies in different regions. Only significant corresponding linear**





regressions are shown in dotted lines with equations and $R^2$ in the same colours. Statistical information for these regressions is shown in Table S1.

We found significant correlations between log(PN) and log(ANF) in the SCS ($r = 0.73$, $p < 0.05$) and ETSP ($r = 0.47$, $p < 0.05$, Fig. 10 *a* and Table S1). Significant correlations between log(SNFR) and log(ANF) were also observed in every compiled study except Mediterranean Sea (Fig. 10 *b* and Table S1). These could be due to non-independent relationships between either PN and ANF, or SNFR and ANF, as N$_2$ fixation rates themselves are derived from PN and SNFR (Equation 2). However, significant correlations were not observed in every compiled study, suggesting that PN concentration is not the

only factor affecting these relationships.

Significant correlation between log(ANF) and log(PN) may also be caused by relatively consistent SNFR or SNFR correlated with PN. The correlation suggests that PN could be a predictive proxy for ANF when detectable. That is, ANF rates could be derived from easily measured PN rather than time and labor consuming N$_2$ fixation rate measurements.

As for the correlation between log(SNFR) and log(ANF), it is noteworthy that the slope varied in different regions (Fig. 10

*b*). Specifically, the ETNP had a higher slope than the others, and WTSP had the lowest. One possible reason is that PN was more quantitatively important in driving N$_2$ fixation rate trends in different areas. On the other hand, no significant correlation is found in the Mediterranean Sea (Table S1), indicating that other controls on ANF need to be further addressed. Interestingly, SCB, Mediterranean Sea and WTSP showed significant correlations between PN concentration and SNFR with varied negative slopes (Fig. 10 *c* and Table S1). This might be due to increasing non-diazotrophs with increasing PN.

Considering PN decreases exponentially with depth, niche separation caused by vertical gradients of environmental factors may also be an underlying reason.

Still, more studies are needed to further indicate global patterns and controlling factors of ANF. It is worth highlighting that complete dataset of PN, SNFR and ANF are essential to discuss their correlations and further examine the applicability of PN as a predictive proxy for ANF.

**4.5 The importance of ANF to the new nitrogen input to the ocean**

The importance of ANF has been raised for less than 20 years due to the long-held idea that the lack of energy source and the high ambient DIN concentration would inhibit the ANF. Our study shows, however, 11 of 18 sampling depths ≥ 200 m had detectable rates, suggesting an overlooked source of N in the ocean's interior.

The depth integrated (200-1000 m) ANF in SCS ranged from 7-86 μmol N m$^{-2}$ d$^{-1}$ (36 ± 26 μmol N m$^{-2}$ d$^{-1}$), contributing 16

± 16 % and 18 ± 24 % of upper 1000 m integrated N$_2$ fixation rates in Kuroshio Current affected and other northern SCS stations, regarding the 0-100 m N$_2$ fixation integration of 463 ± 260 μmol N m$^{-2}$ d$^{-1}$ and 50 ± 19 μmol N m$^{-2}$ d$^{-1}$, respectively (Lu et al., 2019). These are comparable to non-hypoxia studies listed in Table 2. Besides, annual 200-1000 m integrated ANF rate in SCS were calculated to be 13 ± 10 mmol N m$^{-2}$ a$^{-1}$, comparable to estimated atmospheric nitrate and ammonium deposition of around 50 mmol N m$^{-2}$ a$^{-1}$ in northern SCS (Yang et al., 2014). To further estimate annual ANF in SCS, we

calculated the area with bottom depth > 1000 m to be 1.75 × 10$^6$ km$^2$ using GEBCO 2020 data. Using the mean 200-1000 m



integrated ANF of $36 \pm 26$ µmol N m$^{-2}$ d$^{-1}$, the annual ANF in SCS is calculated to be $0.32 \pm 0.24$ Tg N yr$^{-1}$, under the assumption of homogeneous ANF throughout the year. However, particle export in SCS has large seasonal variability with higher particle export in winter (Zhou et al., 2020), which indicates that our annual estimation may be the conservative estimate.

Globally, reported integrated ANF rates ranged from below detection limit to around 600 µmol N m$^{-2}$ d$^{-1}$, contributing up to reported 100 % of water column N$_2$ fixation in Mediterranean Sea (Benavides et al., 2016). Moreover, 8 out of 10 studies and 8 out of 9 studies (including this study) conducted in regions with and without hypoxia reported detectable ANF rates, respectively (Table 2). Quantitatively, mesopelagic N$_2$ fixation would range between 13 and 134 Tg N yr$^{-1}$, significantly influencing global nitrogen budget (Benavides et al., 2018b). Collectively, these suggest that ANF contribute to a
considerable amount of marine N$_2$ fixation.

However, factors controlling ANF are still unclear due to our limited number of observations. As mentioned above, regions with high particle fluxes and high productivity could be the potential hotspots for ANF, such as hypoxic regions (Bonnet et al., 2013; Fernandez et al., 2011; Löscher et al., 2016; Selden et al., 2019) and cold seeps (Dekas et al., 2009; Weber, 2015). Moreover, ANF haven't been explored in many areas, such as subpolar and Polar regions. Only with more comprehensive
datasets can a more precise approximation of global ANF be derived.

**5 Conclusions**

Our comprehensive set of incubations coupled with the persulfate-denitrifier method allow us to provide a solid evidence for ANF in the SCS. To our knowledge, this is the first set of data showing substantial contribution of ANF to new nitrogen inputs in the northwestern Pacific.

High integrated ANF rates corresponded to high sinking particle flux in the SCS, which showed its potential control on ANF. Compiled global ANF data also showed high integrated ANF rates in a highly productive region – ETSP (Bonnet et al., 2013; Fernandez et al., 2011; Löscher et al., 2016), further supporting this hypothesis.

The compilation of global ANF measurements showed detectable and variable rates among different regions, confirming the widespread occurrence of ANF and further suggesting the heterogeneous and complex control of N$_2$ fixation in the aphotic
ocean. In some regions, detectable ANF had significant correlations with PN concentration, suggesting that easily measured PN could be a regional predictive proxy for ANF in the deep ocean.

Based on above trials and results, we here list several recommendations for future ANF studies:

1) In order to detect low ANF rates, we recommend the use of the persulfate-denitrifier method to measure $\delta^{15}$N-PN when sufficient PN mass for the EA-IRMS approach cannot be achieved.

2) Both higher spatial and temporal resolution and corresponding complete datasets of PN, SNFR and ANF are needed to better constrain the controlling factors of ANF and the contribution of ANF to the global N budget.



3) Further rate studies coupled to molecular approach such as nano-SIMS coupled to catalyzed reporter deposition fluorescence in situ hybridization (CARD-FISH) are needed to bridge the knowledge gap between ANF rates and diazotrophs in the deep ocean.

Understanding the magnitudes, mechanisms and limiting factors of ANF not only extends potential niches for $N_2$ fixation, but also better constrains global marine nitrogen cycle.

## Data availability

All the original data are presented in the supporting information and also available from the data share website (at https://doi.org/10.6084/m9.figshare.14178452).

## Author contribution

S.W. and S.-J.K. designed the experiments and S.W. carried them out. S.W., M.D. and S.-J.K. analysed the data. C.S., M.B., S.B., R.H. and C.R.L. provided data for compilation. X.Y. conducted EA-IRMS analyses. S.W. prepared the manuscript draft. X.S.W., C.S., M.B., S.B., R.H., C.R.L., M.R.M. and S.-J.K. contributed to the discussion of the results and revised the manuscript.

## Competing interests

Author Carolin R. Löscher is a member of the editorial board of the journal.

## Acknowledgements

This study was supported by the National Natural Science Foundation of China (NSFC no. 92058204, 41721005). Many thanks to all the cruise member on *R/V Tan Kah Kee* during Aug 2018 and Jul 2017 cruises (KK1806 and KK1905 cruises).
Special thanks to Xirong Chen for his help during the experiment, Xiaowei Chen for his help on flow cytometry data analysis, Zhixiong Huang for EA-IRMS standard data, Wenbin Zou and Li Tian for experimental helps.

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
