# Peer review of "Insights into nitrogen fixation below the euphotic zone: trials in an oligotrophic marginal sea and global compilation"

_Biogeosciences, 2021_

## Referee Comment (RC1)

Abstract:
1) Maybe use biological N2 fixation and abbreviation BNF throughout the abstract and manuscript.
2) Why were amino acids added?
3) Why would the 14N amino acid signal mask 15N2 incubations?
4) Change particulate nitrogen to PN since this abbreviation has already been introduced. Line 33
5) What kind of relationship does PN have with aphotic BNF rates?

Introduction:
1) Include some citations for those studies reporting discrepancies in N budget from bottom-up measurements (line 41).
2) At line 45, consider introducing that the enzyme, nitrogenase, is responsible for BNF and that *nifH*-gene encodes for this enzyme. Take out "and" in "…new N to the open ocean (citation), and…".
3) Line 60: start a new paragraph with "However, owing to the…"
4) Line 68: start a new sentence with "Thus, further investigations…"
5) Line 77: correct spelling for "anaerobic". Also, check out recent publication regarding BNF rates with sinking marine particles (Chakraborty et al., 2021) – very relevant for your research. "Quantifying nitrogen fixation by heterotrophic bacteria in sinking marine particles" Chakraborty et al., 2021
6) Line 81-89: Maybe table 2 could be moved up to the intro and you could incorporate columns with what DOM compound was added and how the BNF rate was influenced in response to the amendment - might enhance the table.
7) Line 95: introduce as South China Sea once before using SCS abbreviation.
8) Line 100: state what different controls you used specifically.

Methods:
1) Fig. 1. Use SCS abbreviation for consistency since it has already been introduced.
2) Fig. 2. Not clear what OMD is referring to.
3) Table 1. Specify cruise dates for K1 and WXS stations. The control/treatment column is not super clear – what are all the other incubations considered? Also, I don't see any specifications for the DFAA treatments here.
4) Section 2.4: line 180 "…representing x % of the total organic carbon…" Not certain how these % are being calculated – total organic carbon measurements of what (e.g., are these citations referencing prior measurements of TOC in the SCS)?
5) Lines 185 – 188 – combine with paragraph starting at line 189.
6) Lines 208-210 combine with previous paragraph.
7) Line 232: are depth integrations done using trapezoidal integration?
8) Table 2 would be more useful in the introduction rather than the long list of citations.

Results:
1) Figs. 3 and 4: Consider reducing the y-axis range so it's easier to discern what's happening with most of the samples. For instances where d15N is greater than 100,

consider using a broken y-axis so that you can highlight both high values and the majority of the data that falls within < 100.

2) OMD depth is referenced several times but haven't specified what the abbreviation stands (Oxygen Minimum Depth?) for.

Discussion:
1) Line 335: citations for low [PN] and low ANF rates in aphotic zone? Combine this paragraph with the next one since it's so short and interconnected.
2) Line 345: citations for characterizing "estuarine environments…ultraoligotrophic waters…polar waters" as "low N2 fixation systems".
3) Line 348 – 349: Some citations highlighting some of these bottle effects would be good.
4) Paragraph 347 – 354: This section could use some extra background information and citations in general.
5) Line 365 - 366: You bring up sinking particles. Are you suggesting that the ANF in deep waters is particle associated? Also, do you have any explanation or prediction as to why that heterogeneity exists? Chakraborty et al. (2021) suggest from their modeling work that there's an ephemeral window for BNF to occur with sinking marine particles. Thus, linking this could give more merit to the explanation that heterogeneity of sinking particles could account for some of the heterogeneity in ANF rates.
6) Lines 377-379: Any speculation to why ANF rates were below detection in this study but above detection in the Mediterranean Sea?
7) Line 380: "Horizontally" refers to "spatially" right?
8) Why were parallel light incubations not carried out in the euphotic zone? This would have allowed you to determine how important diazotrophic activity by NCDs (ANF rates) are compared to cyanobacterial diazotrophic activity (light BNF rates).
9) Line 385 – 386: Is higher sinking flux influenced by seasonal patterns and if so, how might you expect ANF rates to be different if you were to conduct sampling in other seasons (i.e., fall, winter, or spring) than summer.
10) Line 387: You mention larger mean cell diameter and higher PN concentrations – any thoughts on what the community composition might be like (e.g., is it dominated by picoeukaryotes or larger, faster sinking phytoplankton like diatoms)?
11) Line 403: "…or ANF was limited by other factors in the SCS…" could you elaborate? Are there any studies that show DFAA assimilation is preferable to BNF?
12) Line 415: Could you discuss how results might have been different with just a labile C source amendment over a DFAA addition? The DFAA also provides additional labile N to the microbial community but with just a labile C addition, N would be more limiting and could potentially result in a greater stimulation of the diazotrophic community.
13) Line 432: Review Chakraborty et al., 2021 and incorporate their findings in your discussion – it nicely supports the points you are trying to make.
14) Line 462: Could be incorporated as the last sentence for the paragraph rather than a separate paragraph.
15) Line 467-468: "Taken together with previous reports from different oceanographic regions highlighted in Table 2, our study shows 11 of 18 sampling depths…"

16) Line 470: This citation (SEATS might be an overlapping station) is useful for the discussion here and brings up an important point that is currently overlooked: how do these ANF rates compare to BNF rates by cyanobacterial diazotrophs in the SCS? Citation: Nitrogen Fixation by *Trichodesmium* and unicellular diazotrophs in the northern South China Sea and the Kuroshio in summer. Wu et al., 2018 Another citation (with SEATS as overlapping station) that could be useful to compare how ANF rates contribute to net community production at SEATS: Estimated net community production during the summertime at the SEATS time-series study site, northern South China Sea: Implications for nitrogen fixation. Chou et al., 2006

17) Line 477 – 479: How is particle export in the summer compared to other seasons? Could you elaborate more why higher particle export in winter would indicate your annual estimation is conservative? Also, how do your annual ANF contributions to SCS compare to other estimates of BNF contributions to the N budget in SCS (if they exist)?

18) Line 480: How could this be a global interpretation if it's based on one study from the Mediterranean Sea?

19) General discussion needs improvement and more pointed hypothesis or predictions.

Conclusions:

1) Line 495: You didn't have any data suggesting high particle flux was occurring – in fact, you suggest that particle flux is much higher in the winter months. You could more appropriately discuss the potential for higher integrated ANF rates than measured in this study to occur in the winter if seasonal ANF surveys were conducted in the SCS.

2) Paragraphs could be better organized and compiled into fewer paragraphs.

---

## Author Comment (AC1)

**Insights into nitrogen fixation below the euphotic zone: trials in an oligotrophic marginal sea and global compilation**

**Submitted to *Biogeosciences* - Manuscript Number: BG-2021-104**

**Response to reviewer comments**

We would like to appreciate the reviewer for the comments, which are very helpful in improving this manuscript. Below are our point by point responses to the comments.

**Abstract:**

1) Maybe use biological N2 fixation and abbreviation BNF throughout the abstract and manuscript.

Changes made throughout the manuscript except "$N_2$ fixation" from methodological perspective of $^{15}N$-PN enrichment after $^{15}N_2$ addition.

2) Why were amino acids added?

Amino acids were added to examine the effects of labile organic matter on aphotic heterotrophic nitrogen fixation. We changed "distinguish biological $N_2$ fixation." to "distinguish BNF from other biological and non-biological processes and examine the effects of labile organic matter on aphotic BNF" (line 26).

3) Why would the 14N amino acid signal mask 15N2 incubations?

BNF signals are reflected by the increment of $\delta^{15}N$-PN with respect to a time-zero sample. The uptake of $^{14}N$-amino acid can dilute $\delta^{15}N$-PN via the incorporation of $^{14}N$ into particulate matter, thus, may mask BNF signals. We changed "Amino acid additions … added $^{14}N$-amino acid" to "In addition, $N_2$ fixation signals reflected by the increment of $\delta^{15}N$-PN could be masked via isotope dilution due to the incorporation of added $^{14}N$-amino acids into particulate form." (line 27-28).

4) Change particulate nitrogen to PN since this abbreviation has already been introduced. Line 33

Changes made.

5) What kind of relationship does PN have with aphotic BNF rates?

It is the exponential relationship between PN and aphotic BNF rates. We have modified the text to be more specific. We changed "Regression analysis … " (line 33) to "Detectable aphotic BNF rates were exponentially related to PN concentrations in the SCS (n = 11, $p < 0.05$) and eastern tropical south Pacific (n = 33, $p < 0.05$), suggesting the potential of PN concentrations to be a predictive parameter for aphotic BNF".

**Introduction:**

1) Include some citations for those studies reporting discrepancies in N budget from bottom-up measurements (line 41).

We added following citations:

Galloway et al., 2004:

Galloway, J. N., Dentener, F. J., Capone, D. G., Boyer, E. W., Howarth, R. W., Seitzinger, S. P., Asner, G. P., Cleveland, C. C., Green, P. A., Holland, E. A., Karl, D. M., Michaels, A. F., Porter, J. H., Townsend, A. R., and Vöosmarty, C. J.: Nitrogen Cycles: Past, Present, and Future, Biogeochemistry, 70, 153–226, https://doi.org/10.1007/s10533-004-0370-0, 2004.

Großkopf et al., 2012:

Großkopf, T., Mohr, W., Baustian, T., Schunck, H., Gill, D., Kuypers, M. M. M., Lavik, G., Schmitz, R. A., Wallace, D. W. R., and LaRoche, J.: Doubling of marine dinitrogen-fixation rates based on direct measurements, Nature, 488, 361–364, https://doi.org/10.1038/nature11338, 2012.

Xinning Zhang et al., 2020:

Zhang, X., Ward, B. B., and Sigman, D. M.: Global Nitrogen Cycle: Critical Enzymes, Organisms, and Processes for Nitrogen Budgets and Dynamics, Chem. Rev., 120, 5308–5351, https://doi.org/10.1021/acs.chemrev.9b00613, 2020.

2) At line 45, consider introducing that the enzyme, nitrogenase, is responsible for BNF and that nifH-gene encodes for this enzyme. Take out "and" in "...new N to the open ocean (citation), and...".

We added sentence at line 47 and the related citation: "Nitrogenase is the enzyme responsible for BNF, and the highly conserved nitrogenase reductase gene (*nifH*) has been frequently targeted to detect diazotrophs in the ocean (Zehr and Capone, 1996)."

Zehr, J. P., and Capone, D. G.: Problems and promises of assaying the genetic potential for nitrogen fixation in the marine environment, Microbial Ecology, 32(3), 263-281, https://doi.org/10.1007/bf00183062, 1996.

3) Line 60: start a new paragraph with "However, owing to the..."

Changes made.

4) Line 68: start a new sentence with "Thus, further investigations..."

Changes made.

5) Line 77: correct spelling for "anaerobic". Also, check out recent publication regarding BNF rates with sinking marine particles (Chakraborty et al., 2021) – very relevant for your research. "Quantifying nitrogen fixation by heterotrophic bacteria in sinking marine particles" Chakraborty et al., 2021

Correction made and citation added on Line 77 "… microbial consortia (Bombar et al., 2016; Chakraborty et al., 2021; Dekas et al., 2009)".

Chakraborty, S., Andersen, K. H., Visser, A. W., Inomura, K., Follows, M. J., and Riemann, L.: Quantifying nitrogen fixation by heterotrophic bacteria in sinking marine particles, Nat. Commun., 12, 4085, https://doi.org/10.1038/s41467-021-23875-6, 2021.

6) Line 81-89: Maybe table 2 could be moved up to the intro and you could incorporate columns with what DOM compound was added and how the BNF rate was influenced in response to the amendment - might enhance the table.

Table 2 was moved up to the introduction part and changed order with Table 1. Also, DOM addition experiments as well as their effects on $N_2$ fixation were added to the table. In the meantime, we replaced long references in line 81-91 by referring to the table. Finally, the table is updated as below:

**Table 1. Compilation of $N_2$ fixation rate studies conducted below the euphotic zone using $^{15}N_2$ methods. For studies without explicit depth of euphotic zone, only data $\geq 200$ m were included. BDL stands for below detection limit. DOM represents dissolved organic matter, and DFAA, ATP, and TEP represent dissolved free amino acid, adenosine triphosphate, and transparent exopolymeric particle, respectively. DOM addition effect "+", "0", and "–" denote positive effect, no significant effect, and negative effect, respectively.**

| Location | Depth (m) | $N_2$ fixation rate (nmol N $L^{-1}d^{-1}$) | Aphotic zone contribution to total % | Aphotic zone integrated NFR (µmol N $m^{-2}$ $d^{-1}$) | Method | DOM addition and the effect (+/0/–) | Reference |
|---|---|---|---|---|---|---|---|
| *Hypoxic waters* | | | | | | | |
| Southern California Bight | 500, 885 | 0.7 | ca. 30 % (below DCM to 855 m) | 55 | $^{15}N_2$ bubble | | Hamersley et al., 2011 |

| | | | | | | | |
|---|---|---|---|---|---|---|---|
| Eastern Tropical South Pacific | OMZ core (deepest 400 m) | $1.27 \pm 1.2$ | ca. 90 % (deepest level of 1 µmol L$^{-1}$ O$_2$ to 400 m) | $574 \pm 294$ | $^{15}$N$_2$ bubble | | Fernandez et al., 2011 |
| Baltic Sea | 200 | $0.44 \pm 0.26$ | [a]6 % (Suboxic and anoxic area) | Not reported | $^{15}$N$_2$ bubble | | Farnelid et al., 2013 |
| Eastern Tropical South Pacific | 200-2000 | BDL-0.6 | 87-90 % (below the euphotic zone to 2000 m) | 119-501 | $^{15}$N$_2$ bubble | DFAA (+) Carbohydrate (+/0) ATP (0) | Bonnet et al., 2013 |
| Eastern Tropical South Pacific | 200 | 0.37 | Not reported | Not reported | $^{15}$N$_2$ bubble | | Dekaezemacker et al., 2013 |
| Peruvian OMZ | 200 | 0.4 | Not reported | Not reported | $^{15}$N$_2$ bubble | | Loescher et al., 2014 |
| Eastern Tropical South Pacific | 200-500 | BDL-4.39 | Not reported | 150.6-628.7 (0-500 m) | $^{15}$N$_2$ enriched seawater | | Löscher et al., 2016 |
| Pacific Northwest coastal upwelling system | 600 | BDL | Not reported | Not reported | $^{15}$N$_2$ enriched seawater | | Gradoville et al., 2017 |
| Eastern Tropical South Pacific | 200-350 | BDL | Not reported | Not reported | Modified $^{15}$N$_2$ bubble | | Chang et al., 2019 |
| Eastern Tropical North Pacific | 110-3001 | BDL-35.9 | Not reported | Not reported | Modified $^{15}$N$_2$ bubble | DFAA (+/0/–) Carbohydrate (+/0/–) | Selden et al., 2019 |
| *Without hypoxia* | | | | | | | |
| Levantine Basin | 250-500 | 0.01-0.24 | 37-75 % (0.1 % PAR to 500 m) | Not reported | $^{15}$N$_2$ bubble | TEP (+) | Rahav et al., 2013 |
| Gulf of Aqaba | 150-720 | 0.02-0.38 | 56 % (0.1 % light to 720 m) | Not reported | $^{15}$N$_2$ bubble | DFAA (+) | Rahav et al., 2013 |
| Gulf of Aqaba | 200 | 0.2-0.3 | Not reported | Not reported | $^{15}$N$_2$ bubble | DFAA (+) | Rahav et al., 2015 |
| Gulf of Mexico | 330-538 | [b]$1.06 \pm 0.24 * 10^{-5}$ h$^{-1}$ | Not reported | Not reported | $^{15}$N$_2$ bubble | | Weber, 2015 |
| Solomon Seas | 200-1000 | BDL-0.35 | [c]25 % (200-1000 m) | Not reported | $^{15}$N$_2$ enriched seawater | DFAA (+/0) Carbohydrate (+/0/–) | Benavides et al., 2015 |
| Bismarck Seas | 200-1000 | BDL-0.89 | [c]25 % (200-1000 m) | Not reported | $^{15}$N$_2$ enriched seawater | DFAA (+/0) Carbohydrate (-) | Benavides et al., 2015 |
| Mediterranean Sea | 200-2000 | BDL-0.07 | 48-100 % (below 0.01 % PAR to 2000 m) | 17.83-91.06 | $^{15}$N$_2$ bubble | | Benavides et al., 2016 |
| North Pacific Subtropical Gyre | 200 | BDL | Not reported | Not reported | $^{15}$N$_2$ enriched seawater | Carbohydrate (+) | Gradoville et al., 2017 |
| Western Tropical South Pacific | 200-800 | 0.05-0.68 | [d]ca. 6-88 % (200-800 m) | Not reported | $^{15}$N$_2$ bubble | | Benavides et al., 2018 |
| Western North Atlantic Coastal waters | 19.8-40.5 | 1.77-9.12 | Not reported | Not reported | $^{15}$N$_2$ bubble | | Mulholland et al., 2019 |
| South China Sea | 200-4000 | BDL-0.21 | [e]$39 \pm 32$ % (200-1000 m) | $36 \pm 26$ | $^{15}$N$_2$ enriched seawater/$^{15}$N$_2$ bubble | DFAA (+/0/–) | This Study |

[a]Contribution of annual surface water NF in Baltic Sea
[b]Shown in SNFR
[c]Of 5-70 m integration
[d]Of photic zone integration
[e]Of 0-1000 m dark integration

7) Line 95: introduce as South China Sea once before using SCS abbreviation.

Correction made.

8) Line 100: state what different controls you used specifically.

Correction made.

**Methods:**

1) Fig. 1. Use SCS abbreviation for consistency since it has already been introduced.

Changes made.

2) Fig. 2. Not clear what OMD is referring to.

We updated the full explanation of OMD as oxygen minimum depth in the figure caption.

3) Table 1. Specify cruise dates for K1 and WXS stations. The control/treatment column is not super clear – what are all the other incubations considered? Also, I don't see any specifications for the DFAA treatments here.

We added one more line in the Table 1 to clarify the cruise dates for K1 (Aug 2018) and WXS (Jul 2019) and updated the treatment column and depth column to specify the depths of additional treatments as follows:

**Table 1. Experimental treatments in this study. All incubations were in dark conditions. No tracer and HgCl$_2$ denote tracer-free control and killed control with HgCl$_2$, respectively. DFAA represents dissolved free amino acid addition experiments. PC and FLPE denote polycarbonate and fluorinated polyethylene bottles, respectively.**

| Cruise | Station | Depth (m) | Treatment | BNF Method | Replicates | Time Series (hours) | Bottle Type |
|---|---|---|---|---|---|---|---|
| Aug 2018 | K1 | 5, 15, 30, 50, 75, 100 | | $^{15}N_2$-Enriched Seawater | 2 | 0, 24 | 1 L PC |
| | K1 | 200, 300, 740, 1000 | | $^{15}N_2$-Enriched Seawater | 4 | 0, 24 | 4 L FLPE |
| | SEATS | 5, 15, 30, 50, 75, 100 | | $^{15}N_2$-Enriched Seawater | 2 | 24 | 1 L PC |
| | SEATS | 200, 300, 705, 1000, 2000, 3800 | No tracer/HgCl$_2$ | $^{15}N_2$-Enriched Seawater | 4 | 0, 48, 96 | 4 L FLPE |
| Jul 2019 | SS1 | 5, 15, 25, 50, 75, 100 | | $^{15}N_2$ Gas Bubble | 3 | 0, 24 | 1 L PC |
| | SS1 | 120, 150 | | $^{15}N_2$ Gas Bubble | 3 | 0, 24 | 4 L FLPE |
| | SS1 | 200, 700 | DFAA | $^{15}N_2$ Gas Bubble | 3 | 0, 24, 48 | 4 L FLPE |
| | SS1 | 300, 500, 1000, 4000 | | $^{15}N_2$ Gas Bubble | 3 | 0, 24, 48 | 4 L FLPE |
| | WXS | 5 | | $^{15}N_2$ Gas Bubble | 3 | 0, 24, 48 | 4 L FLPE |
| | WXS | 200, 855 | DFAA | $^{15}N_2$ Gas Bubble | 3 | 0, 24, 48 | 4 L FLPE |

4) Section 2.4: line 180 "...representing x % of the total organic carbon..." Not certain how these % are being calculated – total organic carbon measurements of what (e.g., are these citations referencing prior measurements of TOC in the SCS)?

These % of the total organic carbon (TOC) estimations are based on previous studies, as we were not able to measure TOC. We updated the citation to be Wu et al. (2015), a representative study of TOC profiles in the SCS. Also, to clarify the sentence, we changed it into

"…representing ca. 0.4, 0.8, 4, 20, and 40%, respectively, of the total organic carbon (~45 – 50 μM C: Wu et al., 2015).", and the full reference is:

Wu, K., Dai, M., Chen, J., Meng, F., Li, X., Liu, Z., Du, C., Gan, J.: Dissolved organic carbon in the South China Sea and its exchange with the Western Pacific Ocean, Deep Sea Res Part II: Top Stud Oceanogr., 122, 41–51, https://doi.org/10.1016/j.dsr2.2015.06.013, 2015.

5) Lines 185 – 188 – combine with paragraph starting at line 189.

    Change made.

6) Lines 208-210 combine with previous paragraph.

    Change made.

7) Line 232: are depth integrations done using trapezoidal integration?

    Yes, and we made it clear by "Depth integrated $N_2$ fixation rates were calculated by multiplying the average of two adjacent rates by their depth interval (trapezoidal integration)."

8) Table 2 would be more useful in the introduction rather than the long list of citations.

    Changes made as 6) in **Introduction** part above.

**Results:**

1) Figs. 3 and 4: Consider reducing the y-axis range so it's easier to discern what's happening with most of the samples. For instances where d15N is greater than 100, consider using a broken y-axis so that you can highlight both high values and the majority of the data that falls within < 100.

    Following your suggestions, we changed y-axes in Figure 3 and 4 as below for better visualization.

Figure 3:

[Figure]

△ **Tracer-free control**   ✕ **Killed control (+ HgCl$_2$)**   ◯ **N$_2$ fixation**
—— **Minimum acceptable change for detectable rates**

Figure 4:

[Figure]

2) OMD depth is referenced several times but haven't specified what the abbreviation stands (Oxygen Minimum Depth?) for.

Correction made as above in **Methods** 2).

**Discussion:**

1) Line 335: citations for low [PN] and low ANF rates in aphotic zone? Combine this paragraph with the next one since it's so short and interconnected.

Updated as "… low PN concentrations and low ANF rates compared to the surface (Gruber, 2008; Moisander et al., 2017), making …" with citations added in the reference: Gruber, N.: The marine nitrogen cycle: overview and challenges, in: Nitrogen in the marine environment (Second Edition), edited by: Capone, D. G., Bronk, D. A., Mulholland, M. R.,

Carpenter, E. J., Academic Press, 1-50, https://doi.org/10.1016/B978-0-12-372522-6.00001-3, 2008.

2) Line 345: citations for characterizing "estuarine environments...ultraoligotrophic waters...polar waters" as "low N2 fixation systems".

Citation added: "… or even polar waters (Wang et al., 2019), expanding …".

Wang, W.-L., Moore, J. K., Martiny, A. C., and Primeau, F. W.: Convergent estimates of marine nitrogen fixation, Nature, 566, 205–211, https://doi.org/10.1038/s41586-019-0911-2, 2019.

3) Line 348 – 349: Some citations highlighting some of these bottle effects would be good.

Citations added: "… excretion or cell lysis (Calvo-Díaz et al., 2011; Massana et al., 2001)."

Calvo-Díaz, A., Díaz-Pérez, L., Suárez, L. Á., Morán, X. A. G., Teira, E., and Marañón, E.: Decrease in the Autotrophic-to-Heterotrophic Biomass Ratio of Picoplankton in Oligotrophic Marine Waters Due to Bottle Enclosure, Appl. Environ. Microb., 77, 5739–5746, https://doi.org/10.1128/aem.00066-11, 2011.

Massana, R., Pedrós-Alió, C., Casamayor, E. O., and Gasol, J. M.: Changes in marine bacterioplankton phylogenetic composition during incubations designed to measure biogeochemically significant parameters, Limnol. Oceanogr., 46, 1181–1188, https://doi.org/10.4319/lo.2001.46.5.1181, 2001.

4) Paragraph 347 – 354: This section could use some extra background information and citations in general.

We improved the paragraph as following:

"On the other hand, detectable BNF rates require high enough $\Delta\delta^{15}$N-PN to exceed the natural and methodological variance, i.e. the minimum acceptable change (Montoya et al., 1996). There are two ways to methodologically enhance $\Delta\delta^{15}$N-PN. One is to increase incubation time. With a constant BNF rate throughout the incubation, $^{15}$N from BNF would accumulate in PN, so that $\delta^{15}$N-PN increases with time, yielding higher $\Delta\delta^{15}$N-PN. The other way is to increase $A_{N_2}$. With higher enrichment of substrate, BNF would yield PN with higher $^{15}$N proportion, leading to higher $\Delta\delta^{15}$N-PN. However, increasing incubation time, risks increasing bottle effects, such as community shift and seawater chemical change, as well as the movement of the fixed $^{15}$N to other pools via excretion or cell lysis (Calvo-Díaz et al., 2011; Massana et al., 2001). Thus, increasing $A_{N_2}$ is preferable. Both the $^{15}$N$_2$ bubble and the $^{15}$N$_2$ enriched seawater methods were

used to enhance $A_{N_2}$ in this study. Introducing $^{15}N_2$ enriched seawater to achieve $A_{N_2} > 10\%$ requires a large addition of pre-treated filtered water (Mohr et al., 2010), risking dilution of organisms and nutrient and/or trace metal contamination (White et al., 2020). However, $A_{N_2} > 10\%$ can be easily achieved by using the $^{15}N_2$ bubble method without these risks. Nevertheless, incomplete $^{15}N_2$ gas dissolution (Jayakumar et al., 2017; Mohr et al., 2010; White et al., 2020; Wannicke et al., 2018) with the $^{15}N_2$ bubble method (Montoya et al., 1996) may result in underestimation of rates. Here, we minimized the effect of incomplete $^{15}N_2$ gas dissolution with incubation time $\geq 24$ h (Wannicke et al., 2018), thus the results are robust. Even so, we still recommend following studies to use the more precise modified $^{15}N_2$ bubble method (Klawonn et al., 2015). ”

> 5) Line 365 - 366: You bring up sinking particles. Are you suggesting that the ANF in deep waters is particle associated? Also, do you have any explanation or prediction as to why that heterogeneity exists? Chakraborty et al. (2021) suggest from their modeling work that there's an ephemeral window for BNF to occur with sinking marine particles. Thus, linking this could give more merit to the explanation that heterogeneity of sinking particles could account for some of the heterogeneity in ANF rates.

We updated the following explanations in the manuscript:

"… heterogeneity of suspended particles, which could result from both sampling variability and intrinsic particle characteristics. On the one hand, both low concentration and limited sampling volume (1 – 4 L) resulted in heterogeneity between different sampling bottles. This is also supported by Farnelid et al. (2018) that even though the bacterial community composition in bulk particles is consistent throughout time and space, large variations exist in individual particles. On the other hand, sinking particles could also be collected in our samples occasionally, which were reported to be heterogeneous in chemical composition (Martiny et al., 2013) and bacterial community composition (Boeuf et al., 2019). This could be due to the sporadic export events (Boeuf et al., 2019). The larger variances of $\delta^{15}N$-PN in the sample after the incubation compared to T0 time points further demonstrated the heterogeneity of ANF in deep waters (Fig. 3, 4 and S2). This conclusion is further supported by a model study showing a short ephemeral window for BNF in sinking particles, which is also variable due to above heterogeneity of particles (Chakraborty et al., 2021). How to …"

Reference:

Boeuf, D., Edwards, B. R., Eppley, J. M., Hu, S. K., Poff, K. E., Romano, A. E., Caron, D. A., Karl, D. M., and DeLong, E. F.: Biological composition and microbial dynamics of sinking particulate organic matter at abyssal depths in the oligotrophic open ocean, Proc. National Acad. Sci., 116, 201903080, https://doi.org/10.1073/pnas.1903080116, 2019.

Farnelid, H., Turk-Kubo, K., Ploug, H., Ossolinski, J. E., Collins, J. R., Mooy, B. A. S. V., and Zehr, J. P.: Diverse diazotrophs are present on sinking particles in the North Pacific Subtropical Gyre, Isme J., 13, 170–182, https://doi.org/10.1038/s41396-018-0259-x, 2018.

Martiny, A. C., Vrugt, J. A., Primeau, F. W., and Lomas, M. W.: Regional variation in the particulate organic carbon to nitrogen ratio in the surface ocean, Global Biogeochem. Cy., 27, 723–731, https://doi.org/10.1002/gbc.20061, 2013.

6) Lines 377-379: Any speculation to why ANF rates were below detection in this study but above detection in the Mediterranean Sea?

According to Benavides et al. (2016), deep water formation in the Mediterranean Sea brings labile organic matter from the surface to the mesopelagic, which together with the high mesopelagic temperature of this basin (~13ºC), would support ANF activity, by bringing fresh organic matter and stimulating metabolic activities of diazotrophs, respectively.

7) Line 380: "Horizontally" refers to "spatially" right?

Correction made.

8) Why were parallel light incubations not carried out in the euphotic zone? This would have allowed you to determine how important diazotrophic activity by NCDs (ANF rates) are compared to cyanobacterial diazotrophic activity (light BNF rates).

The light incubations were conducted by other colleagues on the cruises and belong to another project. The results remain unpublished yet. More parameters from these two cruises will also be published in the future, but we cannot use them for now. We can further compare our results with those data after their publication.

9) Line 385 – 386: Is higher sinking flux influenced by seasonal patterns and if so, how might you expect ANF rates to be different if you were to conduct sampling in other seasons (i.e., fall, winter, or spring) than summer.

Seasonal variability of sinking particle flux in the South China Sea is generally insignificant with occasionally higher sinking particle fluxes observed in some depths in winter

(Gaye et al., 2009; Kao et al., 2012; Liang, 2008; Yang et al., 2017). Thus, I would expect ANF to be slightly higher in winter with similar rates in spring, summer and fall.

Gaye, B., Wiesner, M. G., and Lahajnar, N.: Nitrogen sources in the South China Sea, as discerned from stable nitrogen isotopic ratios in rivers, sinking particles, and sediments, Mar. Chem., 114, 72–85, https://doi.org/10.1016/j.marchem.2009.04.003, 2009.

Kao, S., Yang, J. T., Liu, K., Dai, M., Chou, W., Lin, H., and Ren, H.: Isotope constraints on particulate nitrogen source and dynamics in the upper water column of the oligotrophic South China Sea, Global Biogeochem. Cy., 26, GB2033, https://doi.org/10.1029/2011gb004091, 2012.

Liang, Y. J.: Depth and temporal variability of organic carbon, total nitrogen and their isotopic compositions of sinking particulate organic matter and POC flux at SEATS time-series station, northern South China Sea [in Chinese with an English abstract], M.S. thesis, National Sun Yat-sen University, Kaohsing, Taiwan, 2008.

Yang, J. T., Kao, S., Dai, M., Yan, X., and Lin, H.: Examining N cycling in the northern South China Sea from N isotopic signals in nitrate and particulate phases, J. Geophys. Res. Biogeosciences, 122, 2118–2136, https://doi.org/10.1002/2016jg003618, 2017.

10) Line 387: You mention larger mean cell diameter and higher PN concentrations – any thoughts on what the community composition might be like (e.g., is it dominated by picoeukaryotes or larger, faster sinking phytoplankton like diatoms)?

We obtained cell diameter via flow cytometry (see Method). Actually, before SYBR Green I staining, we also checked the autofluorescence. There were no significant autofluorescence observed in samples below 200 m, indicating that there were almost no living Prochlorococcus, Synechococcus, nor picoeukaryotes. There were only few Prochlorococcus observed in 200 m samples. Thus, we suggest that bacteria dominate the community composition below 200 m. This is also supported by Boeuf et al. (2019) that bacteria dominated in abyssal sinking particles.

11) Line 403: "...or ANF was limited by other factors in the SCS..." could you elaborate? Are there any studies that show DFAA assimilation is preferable to BNF?

We updated the sentence as "… or ANF was limited by other factors, such as temperature, in the SCS …". Both Selden et al. (2019) and Chakraborty et al. (2021) pointed out that temperature may affect BNF via altering metabolic activities.

Explanation to Line 403 "… the organisms present preferred DFAA assimilation to $N_2$ fixation as a means of getting N.":

1. To our knowledge, there are no studies showing that diazotrophs prefer DFAA assimilation to BNF. However, Chakraborty et al. (2021) employed lower direct cost of amino acid uptake than that of BNF in the model, and BNF occurred after amino acid depletion according to the model.

2. The "organisms" mentioned here refer to all the organisms, not only diazotrophs, but also other heterotrophic bacteria. We are suggesting that at the community level, DFAA assimilation exceeds BNF, masking the $^{15}N$-PN signal from BNF, leading to BNF rates with DFAA addition that are below detection. This is supported by PN increments with DFAA additions (Fig. 7).

12) Line 415: Could you discuss how results might have been different with just a labile C source amendment over a DFAA addition? The DFAA also provides additional labile N to the microbial community but with just a labile C addition, N would be more limiting and could potentially result in a greater stimulation of the diazotrophic community.

We added this discussion in Line 408:

"… DFAA additions. Thus, we suspect that in the SCS, labile organic carbon (glucose, carbohydrate etc.) addition could be a better option than labile organic carbon and nitrogen (DFAA) addition to test whether labile organic matter could stimulate ANF. Labile organic carbon addition only may also result in more N-limiting environment and potentially result in a greater stimulation of the diazotrophic community. However, previous studies reported ANF were stimulated by labile organic carbon to a lesser degree compared to DFAA, possibly due to different substrate preferences or different carbon- or nitrogen-limiting environments (Benavides et al., 2015; Bonnet et al., 2013; Selden et al., 2019)."

13) Line 432: Review Chakraborty et al., 2021 and incorporate their findings in your discussion – it nicely supports the points you are trying to make.

We incorporated this in Line 431: "… energy sources. Taking this into account, a mathematical model study quantified BNF by heterotrophic bacteria in sinking particles, which is determined by polysaccharide and polypeptide concentrations, particle sinking velocity, and surrounding $O_2$ and $NO_3^-$ concentrations (Chakraborty et al., 2021). However, more field studies are needed for model verification and application to global ANF quantification."

14) Line 462: Could be incorporated as the last sentence for the paragraph rather than a separate paragraph.

Changes made.

15) Line 467-468: "Taken together with previous reports from different oceanographic regions highlighted in Table 2, our study shows 11 of 18 sampling depths..."

Changes made as "However, taken together with previous reports from different oceanographic regions highlighted in Table 2, our study shows 11 of 18 sampling depths $\geq 200$ m had detectable rates, suggesting an overlooked source of N in the ocean's interior."

16) Line 470: This citation (SEATS might be an overlapping station) is useful for the discussion here and brings up an important point that is currently overlooked: how do these ANF rates compare to BNF rates by cyanobacterial diazotrophs in the SCS? Citation: Nitrogen Fixation by Trichodesmium and unicellular diazotrophs in the northern South China Sea and the Kuroshio in summer. Wu et al., 2018

Another citation (with SEATS as overlapping station) that could be useful to compare how ANF rates contribute to net community production at SEATS: Estimated net community production during the summertime at the SEATS time-series study site, northern South China Sea: Implications for nitrogen fixation. Chou et al., 2006

We incorporated this citation and updated from Line 469: "The depth integrated (200-1000 m) ANF in SCS ranged from 7-86 μmol N m$^{-2}$ d$^{-1}$ (36 ± 26 μmol N m$^{-2}$ d$^{-1}$), which are comparable to non-hypoxia studies listed in Table 1, contributing 16 ± 16 % and 18 ± 24 % of upper 1000 m integrated BNF rates in Kuroshio Current affected and other northern SCS stations, regarding the 0-100 m BNF integration of 463 ± 260 μmol N m$^{-2}$ d$^{-1}$ and 50 ± 19 μmol N m$^{-2}$ d$^{-1}$, respectively (Lu et al., 2019). Similar euphotic zone integrated BNF rates were also obtained with an average rate from acetylene reduction assay (Wu et al., 2018). Wu et al. (2018) also measured *Trichodesmium*-based BNF rates, while those are extremely low (0.0031 – 0.013 μmol N m$^{-2}$ d$^{-1}$) possibly due to typhoon influence. Thus, euphotic zone BNF is relatively easily affected by weather (light, stratification, temperature, etc.) while ANF is relatively stable with fewer environmental variants, further yielding stable contribution to N budget. Besides, …"

The suggestion for the comparison between ANF and net community production is difficult to conduct since Chou et al. (2006) only calculated this in the mixed-layer. ANF, i.e. BNF below 200 m, only contributes to this net community production when transported upward,

and belongs to the vertical diffusion term in Chou et al. (2006). Thus, we think that such comparison may distract the readers and deviate from our aim of highlighting the importance of ANF in section 4.5.

17) Line 477 – 479: How is particle export in the summer compared to other seasons? Could you elaborate more why higher particle export in winter would indicate your annual estimation is conservative? Also, how do your annual ANF contributions to SCS compare to other estimates of BNF contributions to the N budget in SCS (if they exist)?

We answered seasonal particle export in Discussion-9). We further elaborated this in Line 477: "However, seasonal variability of organic sinking particle flux in the South China Sea is generally insignificant with occasionally higher fluxes observed in some depths in winter (Gaye et al., 2009; Kao et al., 2012; Liang, 2008; Yang et al., 2017). Thus, we would expect slightly higher rates in winter, with relatively stable ANF rates throughout the year, which indicates that our annual estimation may be the conservative estimate."

Chen et al. (2001) reported the nitrogen budget in the South China Sea without BNF and denitrification. It only included riverine input, precipitation, and water exchange with other marine regions. With our ANF estimation of $0.32 \pm 0.24$ Tg N yr$^{-1}$, ANF contributes $1.1 \pm 0.8$ % to the total input of 29 Tg N yr$^{-1}$. The riverine input is 1.4 Tg N yr$^{-1}$, and the precipitation resulted in 0.43 Tg N yr$^{-1}$, which are comparable to our ANF estimation. We would like to update this in Line 479: "This estimation is equivalent to $1.1 \pm 0.8$ % of the total nitrogen input of 29 Tg N yr$^{-1}$ without biological processes, and comparable to the riverine input of 1.4 Tg N yr$^{-1}$ and precipitation input of 0.43 Tg N yr$^{-1}$ (Chen et al., 2001)."

Chen, C.-T. A., Wang, S.-L., Wang, B.-J., and Pai, S.-C.: Nutrient budgets for the South China Sea basin, Mar. Chem., 75, 281–300, https://doi.org/10.1016/s0304-4203(01)00041-x, 2001.

18) Line 480: How could this be a global interpretation if it's based on one study from the Mediterranean Sea?

Here our aim is to give an overview of global reported integrated ANF rates, and we would like to point out that the Mediterranean Sea study is the one with highest ANF contribution to water column integrated rates. To make this clear, we added the reference to Table 1 in the sentence: "Globally, reported integrated ANF rates ranged from below detection limit to around 600 $\mu$mol N m$^{-2}$ d$^{-1}$ (Table 1), contributing up to reported 100 % of water column BNF in Mediterranean Sea (Benavides et al., 2016)."

19) General discussion needs improvement and more pointed hypothesis or predictions.

With above comments, we improved this part with more in-depth discussions, more elaborate words and more references integrated in each section.

**Conclusions:**

1) Line 495: You didn't have any data suggesting high particle flux was occurring – in fact, you suggest that particle flux is much higher in the winter months. You could more appropriately discuss the potential for higher integrated ANF rates than measured in this study to occur in the winter if seasonal ANF surveys were conducted in the SCS.

We did not intend to suggest seasonal variability here. We bring up "high particle flux" here to explain spatial variation of ANF. We would like to elaborate this sentence to be: "Horizontally, high integrated ANF rates corresponded to high sinking particle flux in the SCS, which showed its potential control on ANF."

2) Paragraphs could be better organized and compiled into fewer paragraphs.

We re-organized the conclusion as follows:

"Understanding the magnitudes, mechanisms and limiting factors of ANF not only extends potential niches for BNF, but also better constrains global marine nitrogen cycle. By using persulfate-denitrifier method, we provide solid evidences for ANF in the SCS, which is the first-hand data showing substantial contribution of ANF to new nitrogen inputs in the northwestern Pacific. Horizontally, high depth-integrated ANF rates corresponded to high sinking particle fluxes in the SCS. Compiled global ANF data also showed high depth-integrated ANF rates in a highly productive region – ETSP, further supporting the potential control of sinking particle on ANF rates. In some regions, detectable ANF had significant correlations with PN concentration, suggesting that easily measured PN could be a regional predictive parameter for ANF. However, this correlation is not ubiquitous globally, suggesting the heterogeneous and complex control of ANF.

We here list several recommendations for future ANF studies:

1) In order to detect low ANF rates, we recommend the use of the persulfate-denitrifier method to measure $\delta^{15}$N-PN when sufficient PN mass for the EA-IRMS approach cannot be achieved.

2) Both higher spatial and temporal resolution and corresponding complete datasets of PN, SNFR and ANF are needed to better constrain the controlling factors of ANF and the contribution of ANF to the global N budget.

3) Further rate studies coupled to molecular approach such as nano-SIMS coupled to catalyzed reporter deposition fluorescence in situ hybridization (CARD-FISH) are needed to bridge the knowledge gap between ANF rates and diazotrophs in the deep ocean."